# Distributed Hierarchical Decomposition Framework for Multimodal Timeseries Prediction

**Wei Ye**[†], **Prashant Khanduri**[‡], **Jiangweizhi Peng**[†], **Feng Tian**[†],
**Jun Gao**[∗], **Jie Ding**[†], **Zhi-Li Zhang**[†], **Mingyi Hong**[†]

[†]**University of Minnesota Twin Cities;** [‡]**Wayne State University;** [∗]**Meta Platforms, Inc.**

*ye000094@umn.edu, khanduri.prashant@wayne.edu, peng0504@umn.edu, tianx399@umn.edu, jungao@meta.com, dingj@umn.edu, zhzhang@cs.umn.edu, mhong@umn.edu*

**Reviewed on OpenReview:** *https://openreview.net/forum?id=fFKWs9HslJ*

## Abstract

We consider a distributed time series forecasting problem where multiple distributed nodes each observing a local time series (of potentially different modalities) collaborate to make both *local* and *global* forecasts. This problem is particularly challenging because each node only observes time series generated from a subset of sources, making it challenging to utilize *correlations* among different streams for accurate forecasting; and the data streams observed at each node may represent different modalities, leading to *heterogeneous* computational requirements among nodes. To tackle these challenges, we propose a *hierarchical* learning framework, consisting of multiple *local* models and a *global model*, and provides a suite of efficient training algorithms to achieve high local and global forecasting accuracy. We theoretically establish the convergence of the proposed framework and demonstrate the effectiveness of the proposed approach using several time series forecasting tasks.

## 1 INTRODUCTION

Time series forecasting has diverse applications in various scientific disciplines, such as healthcare (Che et al., 2018), astronomy (Scargle, 1982), finance (Batres-Estrada, 2015), meteorology (Shi et al., 2015), and traffic engineering (Zhang et al., 2017). Historically, regression-based methods have enjoyed widespread popularity for time series forecasting (Box et al., 2015; Toda & Phillips, 1994; Frigola, 2015). However, the recent success of deep learning architectures for sequence learning tasks has led to the development of deep learning-based time series prediction algorithms that outperform the state-of-the-art regression methods (Wu et al., 2020b; Zheng et al., 2020; Lai et al., 2018; Shih et al., 2019; Beltagy et al., 2020).

Despite the success of deep learning architectures for time series forecasting, research on distributed time series forecasting has been scarce. With the advent of distributed data collection in contemporary settings, there is a pressing need for the development of efficient distributed implementations of time series prediction algorithms. However, a notable gap exists in the availability of distributed forecasting modules at a large scale, particularly for scenarios where individual nodes observe only a subset of the entire data stream. In this work, we address a time series forecasting problem wherein multiple distributed nodes collaborate using their local data streams to generate both local (e.g., forecasting local traffic) and global predictions (e.g., predicting aggregated traffic). Only a handful of works have addressed the problem, with the learning algorithms introduced being straightforward adaptations of centralized approaches (Nguyen et al., 2019; Perera et al., 2022; Wang et al., 2022). The existing distributed time series forecasting algorithms fall short of addressing requirements, such as safeguarding data privacy, capability to handle multimodal data, and providing model flexibility. Key challenges in developing a distributed implementation of time series forecasting systems are:
– ❶ Every node captures only a fraction of the time series data, which introduces complexity in the learning process, as effective time series forecasting relies on harnessing the correlations within the data streams (Lai et al., 2018; Wu et al., 2020b; Zheng et al., 2020).

– ❷ The data streams observed at each node may represent a different modality, e.g., the modern Internet of Things (IoT) (Atzori et al., 2010). Making accurate predictions with multimodal data requires each node to choose an appropriate model for its modality, and importantly demands that the information distilled from these heterogeneous models be properly fused.

– ❸ Communication is a major bottleneck in distributed learning (McMahan et al., 2017b), therefore, there is a need to develop algorithms that avoid high-dimensional parameter sharing and reduce privacy risks by avoiding raw data sharing for privacy-sensitive applications at the same time (Aouedi et al., 2022).

– ❹ Current time series forecasting models lack well-established theoretical underpinnings. Analyzing distributed time series forecasting models presents a significant challenge due to the departure from the conventional independent and identically distributed (i.i.d.) gradient assumptions in training, resulting in biased updates during the training process.

It is by no means clear how to design a system that addresses the above-mentioned challenges and provides accurate predictions while being able to: – ① handle multi-modal data by leveraging current advances of the neural network-based time series models; – ② utilize local data while sharing as little information among the nodes as possible while protecting data privacy; – ③ establish theoretical performance guarantees. In this work, we develop a novel framework, distributed vertical hierarchical decomposition (DIVIDE), which addresses the above-mentioned challenges while guaranteeing theoretical performance with non-i.i.d. data samples. The applicability of the developed framework is much beyond time series prediction tasks and can be in fact used to solve general distributed learning problems. Specifically, DIVIDE subsumes a number of popular modern distributed learning systems wherein the feature space of a single data sample is distributed among the distributed nodes, such as the so-called *vertical* federated learning systems (Wei et al., 2022). For example, Webank uses such models for financial risk assessment for their enterprise customers (Cheng et al., 2020). However, our work addresses a significantly more challenging problem since individual data samples (i.e., pieces of the time series data) are correlated, and they may originate from potentially multimodal sources.

**Contributions.** In this work, we develop a novel distributed time series forecasting framework named DIVIDE. Central to our proposed framework is an innovative hierarchical structure, where several local models are constructed to handle data streams observed at a local level, and concurrently, a global model is created to merge the embeddings generated by these local models. Further, a suite of customized and efficient training algorithms has also been developed to ensure that the system generates high-quality forecasting solutions. Specifically, DIVIDE has the following major benefits:

➤ **Flexibility.** It provides local computing nodes with the freedom to select their model architectures, i.e., each local node can choose from a range of available time series prediction models (such as those rooted in RNN or CNN), taking into account factors like their local data modality, available computing resources, and the intricacy of their specific prediction tasks.

➤ **Effective information sharing.** A global model (either hosted at the server or at all the local nodes) is carefully designed to help fuse the (potentially multimodal) information generated from the local models, to best leverage the correlations among all the data streams. Importantly, the proposed algorithms avoid raw data sharing by exploring certain special structures of the loss function so that nodes only exchange low-dimensional embeddings.

➤ **Theoretical guarantees.** We represent the sampling for time series training as a Markov chain (or series) and demonstrate that the proposed algorithms, when applied to train the hierarchical time series forecasting model, converge towards the set of stationary solutions for the associated training problem.

## 2 Preliminaries

We address a distributed learning problem with $K$ nodes. Each node $k \in [K]$, observes a (potentially multimodal) time series $\mathbf{x}_k^n \in \mathbb{R}^{d_k}$, where $n \in \mathbb{N}$ signifies the time index. We define the global time series, observed collectively across all nodes, as $\mathbf{x}^n \in \mathbb{R}^d$, with $d = \sum_{k=1}^K d_k$. The global time series $\mathbf{x}^n$ is constructed as $\mathbf{x}^n := [(\mathbf{x}_1^n)^T, \ldots, (\mathbf{x}_K^n)^T]^T$. We denote a sequence of multi-dimensional time series samples as:

$$\bar{\mathbf{x}}^n := [\mathbf{x}^{n+1}, \mathbf{x}^{n+2}, \ldots, \mathbf{x}^{n+D}] \in \mathbb{R}^{d \times D} \quad \text{for } n \in \mathbb{N} \tag{1}$$

where $D > 0$ is the *time window*; Define the next $\tau$ samples as:

$$\bar{\mathbf{y}}^n := [\mathbf{x}^{n+D+1}, \ldots, \mathbf{x}^{n+D+\tau}] \in \mathbb{R}^{d \times \tau} \quad \text{for } n \in \mathbb{N}, \tag{2}$$

where $\tau$ is the *time horizon* of the prediction. We note that for general prediction problems with streaming data, the labels $\bar{\mathbf{y}}^n$ can represent a general classification or regression task. The global task then is to predict a function of $\bar{\mathbf{y}}^n$ using $\bar{\mathbf{x}}^n$, denoted as $g(\cdot) : \mathbb{R}^{d \times \tau} \to \mathbb{R}^e$ which transforms a vector of data points at a given time to a summary statistics, e.g., for predicting the total traffic across all agents, we have

$$g(\mathbf{y}^n) := \left[ \sum_{k=1}^{K} \sum_{i=1}^{d_k} \mathbf{x}_k^{n+D+1}[i], \cdots, \sum_{k=1}^{K} \sum_{i=1}^{d_k} \mathbf{x}_k^{n+D+\tau}[i] \right].$$

Here we have utilized the notation $(\bar{\mathbf{x}}^n, \bar{\mathbf{y}}^n) \sim \Pi$ for $n \in \mathbb{N}$ to denote the feature label pairs and where $\Pi$ is the underlying distribution that generates the data.

Now, let's transition to the distributed scenario, where each node can only access a subset of the dimensions of the global time series $\mathbf{x}_k^n \in \mathbb{R}^{d_k}$. In this context, leveraging the local observations, we can follow (1) – (2) to define the tuple $(\bar{\mathbf{x}}_k^n, \bar{\mathbf{y}}_k^n)$ as:

$$\bar{\mathbf{x}}_k^n := [\mathbf{x}_k^{n+1}, \ldots, \mathbf{x}_k^{n+D}] \in \mathbb{R}^{d_k \times D}, \tag{3}$$

$$\bar{\mathbf{y}}_k^n := [\mathbf{x}_k^{n+D+1}, \ldots, \mathbf{x}_k^{n+D+\tau}] \in \mathbb{R}^{d_k \times \tau} \tag{4}$$

for $n \in \mathbb{N}$. Furthermore, the specific objective at each local node, $k \in [K]$ is to forecast the local time series or address a local inference problem up to the time horizon of $\tau$ utilizing a time window $D$. As the local forecasting targets are denoted by $\bar{\mathbf{y}}_k^n$'s, we will refer to them as "labels" accessible at node $k$ throughout. In addition, we note that by using the notation of the local observations above, the tuple $(\bar{\mathbf{x}}^n, \bar{\mathbf{y}}^n)$ defined in (1)-(2) can be equivalently written as:

$$\bar{\mathbf{x}}^n := [(\bar{\mathbf{x}}_1^n)^T, \ldots, (\bar{\mathbf{x}}_K^n)^T]^T \in \mathbb{R}^{d \times D},$$

$$\bar{\mathbf{y}}^n := [(\bar{\mathbf{y}}_1^n)^T, \ldots, (\bar{\mathbf{y}}_K^n)^T]^T \in \mathbb{R}^{d \times \tau}$$

for $n \in \mathbb{N}$. This notation will be useful in the subsequent discussion. Overall, the goal of a forecasting algorithm is to learn a mapping from $\bar{\mathbf{x}}^n$ to $g(\bar{\mathbf{y}}^n)$, and/or mappings from each $\bar{\mathbf{x}}_k^n$ to $\bar{\mathbf{y}}_k^n$ for all $k$, by utilizing a set of training samples, so that some loss function is minimized.

## 3 The proposed framework

### 3.1 Problem formulation

In this subsection, we describe the hierarchical model adopted by the proposed DIVIDE framework, as well as its associated training problem.

➤ **Local models.** We assume that each node $k \in [K]$, maintains an independent local machine learning model (e.g., LSTM (Hochreiter & Schmidhuber, 1997) or RNN (Mikolov et al., 2010)) characterized by its parameters $\theta_k \in \Theta_k \subseteq \mathbb{R}^{P_k}$. These models play a key role in conducting local predictive tasks. They take as input the data streams observed locally, and their output is termed the "local embedding", denoted as $f_k(\theta_k; \bar{\mathbf{x}}_k^n)$ for all $n \in \mathbb{N}$ and all nodes $k \in [K]$. Here, $\bar{\mathbf{x}}_k^n$ is defined as in equation (4). Importantly, it's worth noting that these embeddings often possess much lower dimensions compared to $d_k \times \tau$ for each node $k$ in the set $[K]$. Each node is granted the flexibility to select its local model, depending on its available computational resources and the modality of its local data streams, characterized as $(\bar{\mathbf{x}}_k^n, \bar{\mathbf{y}}_k^n)$. For the sake of brevity in our notation, we will subsequently represent $f_k(\theta_k; \bar{\mathbf{x}}_k^n)$ as simply $f_k(\theta_k)$.

➤ **Global models.** Apart from the individual local models, each node (or the server) also maintains a global machine learning model characterized by parameters $\theta_0 \in \Theta_0 \subseteq \mathbb{R}^{P_0}$. This global model serves the purpose of capturing the correlations among different time series in order to improve the prediction accuracy. The inputs for the global model consist of the local embeddings, while its output, termed as the "global embedding", is

denoted as $f_0(\theta_0; f_1, \ldots, f_K)$. This global embedding shares the same dimension as the label to be predicted. For instance, in the context of time series forecasting, the dimension of the global embedding will be $d \times \tau$ since its design is centered around predicting the $d$-dimensional global time series for a time horizon of $\tau$ (as detailed in equations (1) and (2). It's important to note that the global model is introduced to capture interdependencies among diverse time series data and, in doing so, aggregates the local embeddings from each node to learn and represent these correlations.

**Problem.** The overarching objective of the forecasting problem involves acquiring the optimal local parameters for each node while simultaneously optimizing the global model parameters. Defining $\theta \in \Theta$ with $\theta \coloneqq [\theta_0^T, \theta_1^T, \ldots, \theta_K^T]^T$, $\Theta \coloneqq \cup_{i=0}^K \Theta_i \subseteq \mathbb{R}^P$ and $P = \sum_{i=0}^K P_i$. The goal of the forecasting algorithm is to jointly learn these parameters $\theta \in \Theta$ in a distributed manner to minimize a given loss function denoted as $\mathcal{L} : \mathbb{R}^P \to \mathbb{R}$. This is expressed as

$$\min_{\theta \in \Theta} \left\{ \mathcal{L}(\theta) \coloneqq \mathbb{E}_{(\bar{\mathbf{x}}, \bar{\mathbf{y}}) \sim \Pi}[\mathcal{L}(f_0(\theta_0; f_1(\theta_1), .., f_K(\theta_K)); (\bar{\mathbf{x}}, \bar{\mathbf{y}}))] \right\}, \tag{5}$$

Popular choices of $\mathcal{L}(\cdot)$ for time series forecasting problems are the $\ell_2$ and $\ell_1$ losses (Lai et al., 2018). In the subsequent section, we will utilize the specific structure of these loss functions to design a communication-efficient distributed time series forecasting algorithm. We also point out that problem (5) is in general non-convex since the loss function $\mathcal{L}(\cdot)$ is non-convex w.r.t. $\theta \in \Theta$ for many problems of practical interest, e.g., when the local and global models are neural networks (Jain et al., 2017).

*Remark* 1 (**Model Generality**)**.** We point out that although formulation (5) is designed for time series forecasting problems, it is general enough to model standard distributed learning problems where each node has access to only a partial feature originating from a potentially disparate modality, i.e., when the label feature pairs do not originate from a time series. Specifically, our formulation encompasses a majority of distributed learning models, including vertical federated learning frameworks (Wei et al., 2022). However, note that the presence of both the local and the global models, combined with the fact that the data streams can be of potentially disparate modalities, makes our problem significantly more challenging than the ones considered in the past.

Next, we develop the algorithms to solve problem (5). We first describe a vanilla approach that relies on label sharing. Label sharing may be acceptable for some tasks (e.g., classification or regression), particularly when labels are low-dimensional and describe public or non-personal attributes (e.g., weather conditions or device operational states). However, for the time-series prediction problems in Section 2, the labels and features often share common support (see (4)); moreover, in sensitive domains (e.g., medical or personal behavioral data), labels may encode privacy-critical information. Thus, label sharing may be undesirable. To mitigate this issue, we exploit the separability of commonly used time-series loss functions: by Assumption 1, the loss is *separable* across dimensions, enabling a fully distributed implementation without any label sharing.

## 3.2 Prototype algorithm with label sharing

Our objective is to devise a combined optimization and communication strategy to minimize the global loss in equation (5). Specifically, each node strives to learn a local model $\theta_k$, simultaneously acquiring knowledge of a global model, $\theta_0$, which is shared across all nodes or centrally managed by the server. To commence, let's first calculate the stochastic gradients with respect to both the local and global models. Initially, we sample a data point denoted as $(\bar{\mathbf{x}}, \bar{\mathbf{y}}) \sim \Pi$, and compute the stochastic gradients (SG) as:

$$\begin{aligned} \textbf{Local SG: } \nabla_{\theta_k} \mathcal{L}(\theta; (\bar{\mathbf{x}}, \bar{\mathbf{y}})) &= \nabla_{\theta_k} \mathcal{L}(\theta_0, \theta_1, \ldots, \theta_k; (\bar{\mathbf{x}}, \bar{\mathbf{y}})) \\ &= \nabla_{\theta_k} f_k(\theta_k) \, \nabla_{f_k} f_0(\theta_0, f_1(\theta_1), \ldots, f_K(\theta_K)) \nabla_{f_0} \mathcal{L}(f_0(f_1(\theta_1), \ldots, f_K(\theta_K); \theta_0); \bar{\mathbf{y}}), \end{aligned} \tag{6}$$

which follows from the application of the chain rule and the definition of the loss function in (5). Moreover, note that the local models implicitly depend on the local data partitions (see discussion after (5)). Similar to **Local SG**, we compute the **Global SG** as

$$\begin{aligned} \textbf{Global SG: } \nabla_{\theta_0} \mathcal{L}(\theta; (\bar{\mathbf{x}}, \bar{\mathbf{y}})) &= \nabla_{\theta_0} \mathcal{L}(\theta_0, \theta_1, \ldots, \theta_K; (\bar{\mathbf{x}}, \bar{\mathbf{y}})) \\ &= \nabla_{\theta_0} f_0(f_1(\theta_1), \ldots, f_K(\theta_K); \theta_0) \nabla_{f_0} \mathcal{L}(f_0(f_1(\theta_1), \ldots, f_K(\theta_K); \theta_0); \bar{\mathbf{y}}), \end{aligned} \tag{7}$$

---

**Algorithm 1** DIVIDE with label sharing

---

1: **Input**: Rounds $r = \{0, 1, \ldots, R-1\}$, local learning rates: $\{\eta_k^r\}_{k=1}^K$, server learning rate: $\eta_0^r$
2: **Initialize:** Parameters, $\{\theta_0^0, \theta_1^0, \ldots, \theta_K^0\}$
3: **for** $r = 0$ to $R-1$ **do**
4:     Sample $(\bar{\mathbf{x}}_k^r, \bar{\mathbf{y}}_k^r) \sim \Pi$ (see (4)) $\forall k \in [K]$
5:     Share $f_k(\theta_k^r)$ and $\bar{\mathbf{y}}_k^r$, $\forall k$ with server
6:     Compute $\nabla_{\theta_0} \mathcal{L}(\theta^r; (\bar{\mathbf{x}}^r, \bar{\mathbf{y}}^r))$ at server via (7)
7:     Compute $\nabla_{f_k} f_0(\theta_0^r, f_1, .., f_K), \forall k$ at server
8:     `Update:` $\theta_0^{r+1} = \theta_0^r - \eta_0^r \nabla_{\theta_0} \mathcal{L}(\theta^r; (\bar{\mathbf{x}}^r, \bar{\mathbf{y}}^r))$
9:     Receive at each node $\nabla_{f_k} f_0(\theta_0^r, f_1, .., f_K), \nabla_{f_0} \mathcal{L}(f_0(f_1, .., f_K; \theta_0^r); \bar{\mathbf{y}}^r)$ from server
10:    Compute $\nabla_{\theta_k} \mathcal{L}(\theta^r; (\bar{\mathbf{x}}^r, \bar{\mathbf{y}}^r))$ at each node using (6)
11:    `Update:` $\theta_k^{r+1} = \theta_k^r - \eta_k^r \nabla_{\theta_k} \mathcal{L}(\theta^r; (\bar{\mathbf{x}}^r, \bar{\mathbf{y}}^r)) \quad \forall k \in [K]$
12: **end for**

---

which again follows from the application of the chain rule and the definition of the loss function in (5). Next, to implement an efficient algorithm, one would sample $(\bar{\mathbf{x}}, \bar{\mathbf{y}})$ in each round of training and implement an SG-type algorithm using the SG estimates of (6) and (7). In the following, we first present the vanilla version of DIVIDE, which relies on label sharing among nodes.

**A prototype algorithm.** Algorithm 1 describes the steps of a prototype version of the proposed algorithm. At the beginning of each training round, the local nodes share their low-dimensional embeddings as well as their locally observed labels with the server (see Step 5). The server then utilizes the shared information from each node to update the global model using the **Global SG** constructed using (7) in Step 8. To construct the **Local SG** in (6), each node receives the relevant partial gradient from the server in Step 9. The local models are then updated in Step 11. This process continues until convergence. Note that at the end of the training process, each node has access to the model parameters $\theta_k^R$ while the server will have access to $\theta_0^R$. To make the final predictions, each node forwards their local predictions to the server, which then completes the global prediction task.

We highlight key distinctions between DIVIDE and latent "master-node" architectures (Gilmer et al., 2017) as well as other graph-based methods (Zheng et al., 2020; Wu et al., 2020b). While these methods all employ global coordination and message exchange, Gilmer et al. (2017); Zheng et al. (2020); Wu et al. (2020b) operate in a centralized manner, embedding the nodes within a fully centralized graph model where message passing occurs directly between all node pairs. In contrast, DIVIDE operates in a distributed setting: local nodes communicate only with a global model that aggregates and redistributes information without peer-to-peer exchange. As a result, DIVIDE transmits only low-dimensional embeddings, reducing communication overhead, preserving privacy, and improving scalability. Additionally, we note that privacy of DIVIDE can be enhanced with differential privacy (DP) (Dwork et al., 2014), local DP (Duchi et al., 2013), or secure aggregation (Bonawitz et al., 2017) to formally bound information leakage. Additional discussion of related works is provided in Appendix A.

*Remark* 2 (**Limitations**)*.* There are two major limitations of Algorithm 1: – ❶ As briefly discussed earlier, the first limitation is that the local nodes are leaking data. As noted in the implementation of Algorithm 1, the local labels are shared between the local nodes and the server in order to update the global model parameters $\theta_0$. This may be reasonable for some distributed learning tasks (like classification), but not for time series forecasting. – ❷ The correlations between local nodes are not fully utilized, since the algorithm treats each local node independently and equally. However, there are rich correlations between data streams. For example, as observed in (Nguyen et al., 2019), the mobile edge devices (nodes) placed in the vicinity of each other may lead to correlated workload characteristics, thereby leading to correlated data streams. It is well known in the literature that the key to accurate predictions in time series is to utilize correlation among the data streams (Nguyen et al., 2019; Lai et al., 2018; Wu et al., 2020b; Zheng et al., 2020).

Next, we propose an alternate implementation of DIVIDE that addresses the above two limitations.

---

**Algorithm 2** DIVIDE without label sharing

---

1: **Input**: Rounds $r = \{0, 1, \ldots, R-1\}$, local learning rates: $\{\eta_k^r\}_{k=1}^K$, server learning rate: $\eta_0^r$
2: **Initialize:** Parameters, $\{\theta_0^0, \theta_1^0, \ldots, \theta_K^0\}$
3: **for** $r = 0$ to $R-1$ **do**
4:     Sample $(\bar{\mathbf{x}}_k^r, \bar{\mathbf{y}}_k^r) \sim \Pi$ (see (4)) $\forall k \in [K]$
5:     `Share:` Local embeddings $f_k(\theta_k^r)$ $\forall k$ with all nodes via the server
6:     `Compute:` $\nabla \mathcal{L}_{f_0}(f_0(\theta_0^r; f_1, \ldots, f_K); \bar{\mathbf{y}}_k^r)$ at each node and share with all nodes via the server
7:     `Compute:` $\nabla \mathcal{L}_{f_0}(\cdot; \bar{\mathbf{y}}^r) = \sum_{k=1}^K \nabla \mathcal{L}_{f_0}(f_0(\theta_0^r; f_1, ., f_K); \bar{\mathbf{y}}_k^r)$ at each node using Assumption 1
8:     `Compute:` $\nabla_{\theta_k} f_k(\theta_k^r)$, $\nabla_{f_k} f_0(\theta_0^r, f_1, \ldots, f_K)$, and $\nabla_{\theta_0} f_0(f_1, \ldots, f_K; \theta_0^r)$ at each node ((6), (7)).
9:     `Update:` $\theta_0^{r+1} = \theta_0^r - \eta_0^r \nabla_{\theta_0} \mathcal{L}(\theta^r; (\bar{\mathbf{x}}^r, \bar{\mathbf{y}}^r))$
10:    `Update:` $\theta_k^{r+1} = \theta_k^r - \eta_k^r \nabla_{\theta_k} \mathcal{L}(\theta^r; (\bar{\mathbf{x}}^r, \bar{\mathbf{y}}^r))$ $\forall k \in [K]$
11: **end for**

---

### 3.3 Algorithm without label sharing

In this section, we slightly modify the prototype implementation of DIVIDE that allows us to avoid any label sharing. Specifically, to tackle the first limitation, the global model is moved to the local nodes. During the training process, both the local and the global model parameters are updated locally while the server helps orchestrate the communication between the nodes. Note that the global parameters, even though available locally, are still global since they are kept identical at each node throughout the training process. Moreover, we make the following assumption on the loss function.

***Assumption* 1.** $\mathcal{L}(\theta)$ is decomposable across dimensions, i.e., $\mathcal{L}(\cdot\,; (\bar{\mathbf{x}}, \bar{\mathbf{y}})) = \sum_{k=1}^K \mathcal{L}(\cdot\,; (\bar{\mathbf{x}}, \bar{\mathbf{y}}_k))$.

We note that the above assumption holds for many loss functions of interest, including the $\ell_2$- and $\ell_1$-losses that are almost exclusively used for time series prediction problems (Lai et al., 2018). Now, using the above assumption, we can implement the **Local SG** and the **Global SG** without requiring label sharing. The detailed procedure for solving (5) while operating under Assumption 1 is provided in Algorithm 2. It is noteworthy that, following the initial sharing of local embeddings among the nodes (Step 5), each local node computes its partial gradient $\nabla \mathcal{L}(\cdot; \bar{\mathbf{y}}_k^r)$ based on its locally observed data in Step 6. Subsequently, these local partial gradients are shared among nodes to construct the complete partial gradient denoted as $\nabla \mathcal{L}(\cdot; \bar{\mathbf{y}}^r)$ using Assumption 1 in Step 7. Moving forward to Step 8, the remaining partial gradients are calculated using the shared local embeddings to construct both the **Local SG** and the **Global SG** as defined in (6) and (7), respectively. Ultimately, the global and local models are updated concurrently using the computed stochastic gradients in Steps 9 and 10, respectively.

It's important to highlight that the DIVIDE framework offers the freedom to select both the global and local models. The local models are responsible for capturing the specifics of the individual time series data observed locally, while the global models are designed to uncover the correlations between the distinct time series observed across different nodes. Next, we discuss the communication required by DIVIDE.

***Remark* 3 (Communication).** Algorithm 1 requires information sharing in two steps, namely Steps 5 and 9. Note that in Step 5, the local embeddings and labels are shared with the server, while in Step 9, the local (and global) intermediate partial gradients are shared. As discussed earlier in Section 3, the local embeddings are low-dimensional mappings (with dimensions much smaller than $d_k \times \tau$ for each $k \in [K]$) while the global embeddings have dimension $d \times \tau$. This implies that the total network-wide communication required by DIVIDE in each round of communication is only $4d\tau$. Similarly, in Algorithm 2, the information sharing is conducted in Steps 5 and 6. In Step 5, the local embeddings are shared (back and forth) among all the nodes, while in Step 6, the decomposed local gradients (across dimensions) are shared among the nodes, which are then utilized to construct the full local and global SG. In Step 5, a total of $2d\tau$ real values are shared (back and forth) among nodes. Similarly, in Step 6, a total of $2Kd\tau$ real values are shared, making the total communication of $2d\tau + 2Kd\tau$.

Compared to standard federated learning methods, such as FedAvg (McMahan et al., 2017a), FedProx (Li et al., 2020), FedDyn (Acar et al., 2021), FedNova (Wang et al., 2020), which synchronize full model

parameters of size $P$ each round at $O(KP)$ cost, DIVIDE only exchanges low-dimensional embeddings, achieving significantly lower cost since $d \ll P$. While compression-based federated learning variants, like QFedAvg (Li et al., 2019b), SignSGD (Bernstein et al., 2018), TernGrad (Wen et al., 2017), or sparsified federated learning techniques (Aji & Heafield, 2017) reduce communication cost, they still fundamentally scale with $P$ due to increase in gradient variance (Shao et al., 2023; Han et al., 2020).

## 4 Convergence guarantees

In this section, we provide the convergence analysis of the proposed approach. First, we note that the training samples $(\bar{\mathbf{x}}^n, \bar{\mathbf{y}}^n)$ for $n \in \mathbb{N}$ utilized for solving (5) are non i.i.d. (see (1) and (2)). This follows from the fact that each pair of consecutive training samples $\bar{\mathbf{x}}^{i-1}$ and $\bar{\mathbf{x}}^i$ for $i \in \mathbb{N}$ share some common time series observations, i.e. $\bar{\mathbf{x}}^i := [\mathbf{x}^{i+1}, \dots, \mathbf{x}^{i+D}]$ shares the first $D-1$ observations with $\bar{\mathbf{x}}^{i-1} := [\mathbf{x}^i, \dots, \mathbf{x}^{n+D-1}]$. This implies that the stochastic gradients computed using these non-i.i.d. samples in Algorithms 1 and 2 will be biased and may cause the algorithms to diverge. However, note that the samples $(\bar{\mathbf{x}}^n, \bar{\mathbf{y}}^n)$ follow the Markov property, meaning that we have $\mathbb{P}[\bar{\mathbf{x}}^i | \bar{\mathbf{x}}^{i-1}, \dots, \bar{\mathbf{x}}^0] = \mathbb{P}[\bar{\mathbf{x}}^i | \bar{\mathbf{x}}^{i-1}]$, which can be directly observed from the definition of $\bar{\mathbf{x}}^i$ in (1). Specifically, the Markov property follows since conditioning on $\bar{\mathbf{x}}^{i-1}$ the only new sample that is observed is $\mathbf{x}^{i+D}$ while the rest of the samples have already been observed in $\bar{\mathbf{x}}^{i-1}$. Motivated by this observation, we make the following assumptions about the data-generating process.

***Assumption 2.*** We assume that the data-generating process $\{\bar{\mathbf{x}}^n, \bar{\mathbf{y}}^n\}_{n \geq 0}$ follows a Markov chain trajectory with $M$ states. The Markov chain is time-homogeneous, irreducible, and aperiodic. The Markov chain has a transition matrix $T \in \mathbb{R}^{M \times M}$ and stationary distribution $\Pi^*$.

To the best of our knowledge, convergence analyses that explicitly exploit the Markov property of time-series data in distributed forecasting settings remain scarce and largely open. We also point out that Assumption 2 is also a practical assumption for other classes of problems (classification/regression/reinforcement learning) since for many cases it is easy to obtain samples from Markov chain trajectories rather than obtaining i.i.d. samples (Sun et al., 2018; Doan et al., 2020). Also, note that Assumption 2 assumes that the samples are generated from a finite state space. This assumption can easily be relaxed when the data samples $\{\bar{\mathbf{x}}^n, \bar{\mathbf{y}}^n\}_{n \geq 0}$ are generated from a Markov series rather than a Markov chain (Sun et al., 2018). Next, we make some assumptions about the loss function $\mathcal{L}(\cdot; (\bar{\mathbf{x}}, \bar{\mathbf{y}}))$.

***Assumption 3.*** The **Local SG** and the **Global SG** derived in (6) and (7) are bounded, i.e., we have $\|\nabla_{\theta_k} \mathcal{L}(\theta; (\bar{\mathbf{x}}, \bar{\mathbf{y}}))\| \leq G$ and $\|\nabla_{\theta_0} \mathcal{L}(\theta; (\bar{\mathbf{x}}, \bar{\mathbf{y}}))\| \leq G$. We also assume that the loss function is $L$-Lipschitz smooth, i.e., $\nabla_\theta \mathcal{L}(\theta; (\bar{\mathbf{x}}, \bar{\mathbf{y}}))$ is $L$-Lipschitz.

Assumption 3 is a standard assumption in first-order algorithm analyses and has also been made in earlier works (Sun et al., 2018). Next, we state the convergence performance of DIVIDE.

**Theorem 4.1.** *Suppose Assumptions 2 and 3 hold, and that the learning rates satisfy:*

$$\sum_r \eta_k^r = +\infty, \ \sum_r \ln^2 r \cdot (\eta_k^r)^2 < +\infty \ \ \forall k \in \{0, 1, \dots, K\}.$$

*Then we have $\lim_{R \to \infty} \mathbb{E}\|\nabla_\theta \mathcal{L}(\theta)\| = 0$, where $\mathcal{L}(\theta)$ is defined in (5). Moreover, we have*

$$\mathbb{E}\Big[ \min_{1 \leq r \leq R} \{\|\nabla_\theta \mathcal{L}(\theta^r)\|^2\} \Big] = \mathcal{O}\left( \frac{\Psi(T)}{\sum_{r=1}^R \min\{\eta_k^r\}_{k=0}^K} \right),$$

*where $\theta^r := [(\theta_0^r)^T, \dots, (\theta_K^r)^T]^T$ and $\Psi(T)$ is*

$$\Psi(T) := \max\left\{ 1, \frac{1}{\ln(1/\lambda(T))} \right\},$$

*where $\lambda(T) := \frac{\max\{|\lambda_2(T)|, |\lambda_M(T)|\} + 1}{2} \in [0, 1)$ and $\lambda_i(T) \in \mathbb{C}$ is the $i^{th}$ largest eigenvalue of $T$.*

We note that the above result matches the guarantees of a centralized Markov Chain Gradient Descent for $K = 1$ (Sun et al., 2018). Moreover, distributed Markov chain gradient descent under different settings have

been considered in the past (Sun & Li, 2019; Wai, 2020). Specifically, the setting in Sun & Li (2019); Wai (2020) assumes that each node has access to a complete data sample, however, in the DIVIDE framework, each node only observes a partial time series. Also, in Sun & Li (2019); Wai (2020) each node learns the same model, and therefore, has the same local step sizes across the network. In contrast, in DIVIDE framework each node can flexibly choose the local model based on its local modality and learning task. Consequently, each node can perform local learning utilizing different step sizes. In addition, the DIVIDE framework allows each node to maintain a global model to learn the correlations among different time series. These key distinctions make the analysis of DIVIDE significantly challenging compared to prior works. Theorem 4.1 implies the following.

**Corollary 1.** If we choose the learning rates $\eta^r = \mathcal{O}(1/r^q)$ with $q \in (1/2, 1)$, then DIVIDE achieves

$$
\mathbb{E}\Big[\min_{1 \le r \le R}\{\|\nabla_\theta \mathcal{L}(\theta^r)\|^2\}\Big] = \mathcal{O}\left(\frac{\Psi(T)}{R^{1-q}}\right).
$$

It is important to note that the non-convexity of the loss function $\mathcal{L}(\theta)$ (defined in (5)) suggests that we cannot expect a gradient-based algorithm to converge to the globally optimal solution. Theorem 4.1 and Corollary 1 establish the convergence of DIVIDE to a stationary point, in expectation.

## 5 Numerical experiments

In this section, we evaluate the performance of the proposed algorithm on real-world datasets with a number of time-series forecasting baselines implemented in both centralized and distributed manners.

### 5.1 Experiment setup

We describe our numerical experiment setup below and refer the readers to Appendix B for more information about implementation details, hyperparameter settings, training and test schemes, etc.

**Datasets.** Our evaluation is conducted on a number of diverse real-world datasets: [**D1**] Power demand dataset (POWR) (EIA, 2022), which contains the hourly power demand of thirteen major electricity grid service regions in the United States for 2022; [**D2**] NYC subway traffic dataset (SUBW) (EDDEN, 2021), which collects the number of people entering and exiting each subway station per hour in NYC, 2017 – 2021; [**D3**] Meteorological dataset (WEAT) (BENIAGUEV, 2017), which reports weather data from 36 regions with different indicators from Oct. 2012 to Nov. 2017. [**D4**] DeepSense6G (DS6G) (Charan et al., 2022), which is a multimodal dataset containing sensory (e.g., camera, Radar, LiDAR) and radio information. Same with the ITU 2022 challenge (ITU, 2022), we use four scenarios in experiments. Those datasets all inherently handle distributed scenarios as they are collected from independent sources.

**Baseline models.** We adopt five widely used deep learning baselines: LSTM (Hochreiter & Schmidhuber, 1997), TCN (Bai et al., 2018), LSTNet (Lai et al., 2018), Informer (Zhou et al., 2021), and DLinear (Zeng et al., 2023). These baselines are implemented in both centralized and distributed fashions. When incorporated as local models within our distributed framework. We denote them as "DIVIDE (local model)". For example, DIVIDE (LSTM) adopts LSTM in the local nodes. Additionally, we also consider the Prophet (Taylor & Letham, 2018b) as the baseline, which is a widely recognized statistics-based algorithm. For the multivariate time-series forecasting task, we optimize the "multi-prophet" library (Keča, 2020), where independent prophet models will model each data stream.

**Evaluation metrics.** For the multivariate time-series forecasting task, we utilize root mean square error (RMSE), which gauges the disparity within Euclidean space. For the multimodal time-series prediction with the DeepSense 6G dataset, we use the distance-based accuracy (DBA) score; see the appendix for a detailed illustration.

Table 1: Comparison of DIVIDE with other baselines for mid/long-term forecasting. The experiments are conducted for both centralized (i.e., LSTM, TCN, LSTNet, INFORMER, and DLinear) and distributed setups (i.e., DIVIDE(LSTM), DIVIDE(TCN), DIVIDE(LSTNet), DIVIDE(INFORMER), and DIVIDE(DLinear)). These methods utilize a 36-hour window to predict horizons (24h, 48h, 72h, 168h). Performance is evaluated via *RMSE*, where *smaller* values indicate superior performance.

| Models | | Multi-Prophet | LSTM | DIVIDE (LSTM) | TCN | DIVIDE (TCN) | LSTNet | DIVIDE (LSTNet) | INFORMER | DIVIDE (INFORMER) | DLinear | DIVIDE (DLinear) |
|---|---|---|---|---|---|---|---|---|---|---|---|---|
| POWR | 24h | 0.0764 | 0.0313 | 0.0303 | 0.0349 | 0.0321 | 0.0375 | 0.0374 | 0.0245 | 0.0268 | **0.0244** | 0.0249 |
| | 48h | 0.0787 | 0.0403 | 0.0411 | 0.0446 | 0.0439 | 0.0449 | 0.0427 | 0.0289 | 0.0303 | **0.0287** | 0.0312 |
| | 72h | 0.0791 | 0.0475 | 0.0447 | 0.0498 | 0.0479 | 0.0501 | 0.0468 | **0.0315** | 0.0334 | 0.0329 | 0.0325 |
| | 168h | 0.0820 | 0.0574 | 0.0575 | 0.0583 | 0.0546 | 0.0608 | 0.0588 | 0.0410 | **0.0402** | 0.0425 | 0.0437 |
| SUBW | 24h | 0.1242 | 0.0816 | 0.0765 | 0.0875 | 0.0941 | 0.0853 | 0.0850 | 0.0609 | 0.0613 | **0.0600** | 0.0623 |
| | 48h | 0.1361 | 0.0823 | 0.0804 | 0.0886 | 0.1040 | 0.0850 | 0.0856 | **0.0657** | 0.0668 | 0.0669 | 0.0689 |
| | 72h | 0.1440 | 0.0844 | 0.0850 | 0.0865 | 0.1203 | 0.0868 | 0.0865 | 0.0712 | 0.0720 | **0.0710** | 0.0726 |
| | 168h | 0.1452 | 0.0912 | 0.0892 | 0.0882 | 0.1068 | 0.0925 | 0.0919 | **0.0746** | 0.0785 | 0.0780 | 0.0799 |
| WEAT | 24h | 0.0204 | 0.0102 | 0.0119 | 0.0093 | 0.0118 | 0.0111 | 0.0142 | **0.0086** | 0.0092 | 0.0092 | 0.0089 |
| | 48h | 0.0211 | 0.0107 | 0.0103 | 0.0106 | 0.0126 | 0.0123 | 0.0164 | **0.0088** | 0.0089 | 0.0142 | 0.0137 |
| | 72h | 0.0215 | 0.0133 | 0.0130 | 0.0112 | 0.0113 | 0.0126 | 0.0137 | 0.0119 | **0.0108** | 0.0154 | 0.0140 |
| | 168h | 0.0246 | 0.0125 | 0.0136 | 0.0139 | 0.0146 | 0.0222 | 0.0264 | **0.0124** | 0.0135 | 0.0159 | 0.0153 |

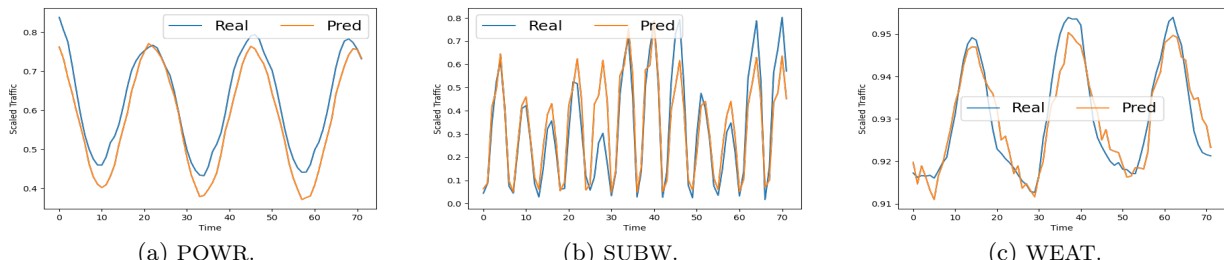

| (a) POWR. | (b) SUBW. | (c) WEAT. |
|---|---|---|

Figure 1: Visualized forecast results of future 72-hrs using 36-hrs historical window via DIVIDE(TCN).

## 5.2 Distributed time-series forecasting

Following the above experiment setups, we now demonstrate the validity of DIVIDE by comparing it with baseline algorithms. Unless otherwise specified, we configure eight local nodes, some of which might include multiple data streams. The results are reported in Table 1.

**Comparison with multi-Prophet.** We see that DIVIDE significantly outperforms Multi-Prophet, likely because DIVIDE employs neural networks and leverages correlations among data streams, thereby enhancing its learning capabilities in contrast to the statistical-based Prophet, which independently models each data stream.

**Comparison with centralized models.** Despite each local node lacking direct access to all data streams under the distributed implementation setting, the results indicate that DIVIDE delivers competitive performance. In certain cases, we even observe that DIVIDE surpasses the centralized baseline. We attribute this performance gain to DIVIDE's ability to efficiently leverage cross-node correlations through structured aggregation, which is also partially supported by the observation (see Sec. 5.3 and Appendix C.2) that models incorporating a global component consistently outperform those without, implying that aggregation mitigates overfitting to local patterns. Moreover, as discussed in Sec 3 and 4, DIVIDE is particularly advantageous under heterogeneous data distributions, where embeddings from different modalities provide complementary information that centralized models may overfit to and fail to fully exploit. Conversely, the dataset's spatio-temporal nature amplifies the training complexity of centralized models. For example, while a weather station in New Jersey may aid in predicting weather patterns in New York, its usefulness for predicting weather in California may be limited. In fact, such characteristics have motivated some centralized time-series prediction models to adopt hierarchical or localized aggregation strategies (Wang et al., 2022; Li et al., 2017).

### 5.3 In-depth discussion on distributed multivariate time-series forecasting

To further examine DIVIDE, we conduct comprehensive studies under various settings and highlight results. We provide more details and additional experiments in Appendix C, including hierarchical prediction tasks, a complete set of ablation studies, and scalability of the proposed algorithms.

**Flexibility of the local models.** As noted before, DIVIDE enables individual nodes to independently select their local models. Table 1 presents the performance of DIVIDE utilizing different types of neural networks as the local model. It is evident that the various local models attain differing levels of performance on different datasets, possibly attributed to the distinctive characteristics of the local model and the datasets. It is also worth noting that DIVIDE also supports the mixed types of local models.

**Impact of the global model.** We evaluate the performance of DIVIDE with and without the MLP global model. As illustrated in Fig. 2, the results clearly demonstrate that incorporating global models enhances forecasting performance by leveraging shared local node information. The numerical results reveal that adopting a global model can lead to about 4.4% performance improvement on average. Meanwhile, we also find that the inclusion of a global model enhances the scalability of DIVIDE, as its performance remains stable regardless of the number of nodes. Conversely, not having a global model results in significantly poorer outcomes as the number of nodes increases; see Appendix C.2 for details.

**Hierarchical tasks.** In addition to the forecasting task on each local node, the design DIVIDE also supports global tasks. In the Appendix C.2, we introduce two global tasks and refer to them as *Sum* that aims to capture an algebraic correlation (i.e., sum) among all local nodes data stream, and *Spatial* that aims to predict one of the local node information based on the others. From Table 5 and Table 6, we see that achieving good performance is challenging for both Prophet and DIVIDE

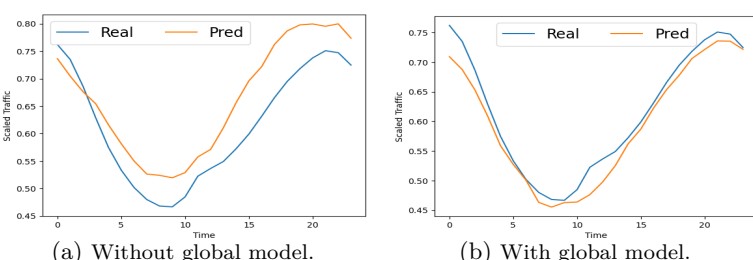

(a) Without global model.      (b) With global model.

Figure 2: 24-hrs forecast of DIVIDE(LSTNet) on POWR dataset with and without global model.

without the global model. It indicates the adaptability of DIVIDE to various tasks.

**Asynchronous model updates.** The local and global models in DIVIDE can be updated asynchronously; see the results attached in Appendix C.2, where we employ different learning rates. The results illustrate that our framework can effectively operate on various configurations and exhibits robust performance, corroborating the earlier findings.

### 5.4 Multimodal time-series prediction

We now utilize DIVIDE for multimodal time-series prediction. In this context, the dataset includes not only numerical time series but also diverse data modalities, such as RGB videos, point cloud streaming, etc. We conduct numerical experiments on the DeepSense6G dataset from the ITU 2022 challenges (Charan et al., 2022; ITU, 2022), aiming to predict the strongest signal power radio beam ID based on the environment sensoring information. To accommodate the different data modalities, we employ distinct local models, and the specific model architectures are outlined in Appendix B.3. We highlight results in Table 2, with more details and discussions in Appendix C.3.

From Table 2, we observe that the global model is able to effectively fuse the information collected from the local model and improve performance by increasing the DBA score up to 15.9% by the single modality. What's more, DIVIDE surprisingly ranks in the Top 5, compared with other solutions on the leaderboard, which reports a total of 150+ submissions (ITU, 2022). It shows DIVIDE 's powerful learning capability in fusing the different modalities. It is worth noting that, since we have not targeted achieving the best result for this specific challenge, but rather to validate the effectiveness of the proposed algorithm in fusing different time-series data modalities, we have not applied comprehensive data preprocessing as other submissions have done for the challenge.

Table 2: The performance achieved by DIVIDE with different input data modalities. Here, Sce. refers to the different scenarios used in the ITU 2022 challenge (ITU, 2022). We use the DBA score as the metric, where *larger* values indicate better performance. DIVIDE ranks in the top 5 among 150+ submissions based on the overall score.

|  | Sce. 31 | Sce. 32 | Sce. 33 | Sce. 34 | Overall |
|---|---|---|---|---|---|
| **GPS-only** | 0.0256 | 0.6518 | 0.6332 | 0.6264 | 0.4843 |
| **Vision-only** | 0.1142 | 0.6163 | 0.6830 | 0.6619 | 0.5189 |
| **LiDAR-only** | 0.0653 | 0.6385 | 0.6120 | 0.5967 | 0.4781 |
| **DIVIDE(All)** | 0.0715 | 0.7450 | 0.6895 | 0.7385 | **0.5611** |

## 6  Conclusion

In this work, we developed a distributed time series forecasting framework that decomposes the entire time series prediction model into multiple local models, which are then composed to construct the global model. This decomposition gives each local and global node the flexibility of the model construction so that they can choose from a variety of existing time series prediction models depending on the local compute resources and the modality of local data. Also, the proposed framework is communication-efficient and mitigates privacy risks by avoiding raw data sharing. We evaluated the proposed framework on a number of time-series forecasting problems and showed that the framework performs well on a majority of multivariate time-series forecasting tasks.

### Acknowledgements

This research is partially funded by Meta Platforms, Inc. In addition, Hong acknowledges support from the USDA National Institute of Food and Agriculture (NIFA) and the National Science Foundation (NSF) National AI Research Institutes Competitive Award (No. 2023-67021-39829), as well as NSF Grants EPCN-2311007 and CNS-2003033. Zhang's contribution is supported in part by NSF Grants CNS-2212318, CNS-2220286, and CNS-2220292. Ding also got supported by the Army Research Office through an Early Career Program Award (No. W911NF-23-10315).

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

## A Related work

**Centralized models.** Time series forecasting has been a problem of interest for decades. Popular statistical approaches for time series prediction include vector autoregression (VAR) (Box et al., 2015), vector auto-regressive moving average model (VARMA) (Toda & Phillips, 1994), and Gaussian processes (GP) (Frigola, 2015). Moreover, the Prophet model developed in Taylor & Letham (2018a) proposes a modular regression model for time series forecasting that relies on decomposing the time series into three main model components, namely, trend, seasonality, and holidays, which are then learned through various statistical learning techniques (Harvey & Shephard, 1993; Rockwood, 2015). These statistical approaches learn interpretable models, but sometimes impose strong assumptions (such as stationarity and linear dependence among variables) on the multiple time series models. To circumvent these issues, recently, deep learning-based models have gained popularity for time series forecasting (Lim & Zohren, 2021). There are many neural network models one can utilize for time series forecasting, however, in practice, three types of neural network architectures, namely Recurrent Neural Networks (RNNs) (Mikolov et al., 2010), Graph neural networks (GNNs) (Wu et al., 2020a), and more recently, transformers (Vaswani et al., 2017) have gained popularity for solving such problems. The works (Lai et al., 2018) and (Shih et al., 2019) propose to utilize RNNs in conjunction with CNNs for time series prediction. The idea behind utilizing such a structure is that the CNNs will capture the spatial dependencies among variables while the RNNs will capture long-term temporal dependencies among the variables. More recently, GNN (Zheng et al., 2020; Wu et al., 2020b) and Transformers-based architectures (Beltagy et al., 2020; Li et al., 2019a; Child et al., 2019; Zhou et al., 2021) have gained popularity for time series forecasting problems. In spite of a number of works on improving the time series forecasting capabilities of different models, there has been a lack of research on distributed implementations of time series forecasting algorithms.

**Distributed time series forecasting.** In the literature, distributed time series forecasting has received relatively less attention as compared to its centralized counterpart, and this is partially because of several technical challenges. To conduct time series forecasting, practitioners usually have to adopt inadequate distributed platforms (Meng et al., 2016; Galicia et al., 2018) (please see discussion in (Wang et al., 2022, Section 1)). In Nguyen et al. (2019), the authors proposed a distributed workload forecasting algorithm for mobile edge devices wherein the authors leveraged correlation among workloads of edge devices in a close physical distance and applied LSTM to perform the prediction. In Perera et al. (2022), the authors proposed distributed forecasting algorithms for solar photovoltaic (PV) power generation, where a certain particle swarm optimization-based approach is used to combine five state-of-the-art forecast models for solar PV power generation prediction. However, a major drawback of the algorithms proposed in Nguyen et al. (2019); Perera et al. (2022) is that they assume that the availability of *the entire* set of data streams is available at a central location, which effectively defeats the purpose of distributed processing. Recently, Wang et al. (2022) proposed a distributed ARIMA architecture for time series forecasting where the authors utilized a map-reduce architecture to perform predictions. In contrast to our model where each node observes a time series from a different source, in (Wang et al., 2022), the authors adopt a model where each node observes *the same* time series but during separate time windows which is an impractical setting compared to the general setting considered in our work. In addition, instead of fixing a local model, our framework endows each

node with the flexibility of choosing a local model best suited for its data modality and available computing resources, while Wang et al. (2022) constraints the individual nodes to utilize ARIMA models (Shumway et al., 2017).

## B    Implementation details

### B.1    Datasets and tasks

**Multivariate time-series forecasting.**    We evaluate the performance of DIVIDE on the three real-world datasets: the Power Demand dataset (POWR) (EIA, 2022), the NYC Subway Traffic dataset (SUBW) (EDDEN, 2021), and the Meteorological dataset (WEAT) (BENIAGUEV, 2017). In the following table, we provide a brief overview of the statistical information of these datasets:

Table 3: Statistical information of the multivariate time-series forecasting datasets.

| Name | #Timestamps | Granularity | #Streams | Physical meanings |
|------|-------------|-------------|----------|-------------------|
| POWR | 8737 | 1 hr | 13 | Each stream contains the electricity demand of one of the 13 major US regions. |
| SUBW | 9,911 | 4 hrs | 469 | Each stream contains the number of passengers entering one subway station in New York City. |
| WEAT | 45,252 | 1 hr | 36 | Each stream contains the temperature information of selected cities; 30 are in North America, and 6 are in middle Asia. |

**Multimodal prediction.**    We also use the DeepSense6G (DS6G) (Charan et al., 2022) to evaluate the performance of DIVIDE for the multimodal prediction task, where we are supposed to utilize the environment information captured by sensors (e.g., camera, Radar, LiDAR) to predict the radio beam with the strongest signal strength. Specifically, we follow the ITU 2022 AI challenge (ITU, 2022) and focus on scenario 31∼34 of DeepSense6G to have a fair competition with the method on the leaderboard. Each scenario is collected at different times and locations.

### B.2    Environment setups

**Softwares assets.**    Our codes mainly rely on the following third-party libraries and all of them can be easily installed: Numpy[1], scikit-learn[2], Pytorch[3], PyG[4], multi-prophet[5], DTAIDistance[6], Pandas[7], Matplotlib[8].

**Hardwares.**    Our workstation runs on the Unbuntu 18 system and is equipped with AMD Ryzen Threadripper PRO 3995WX CPU, 1TB memory, and three Nvidia RTX A6000 GPUs (48GB memory each).

**Hyperparameters Settings.**    For all the experiments of neural network(NN)-based models, we split the datasets into the training, validation, and test sets with the ratio of 0.5 : 0.2 : 0.3. The min-max scaler is applied for the data normalization. All models are trained using the Adam (Kingma & Ba, 2014) optimizer with a batch size of 128. If there is no additional description, we follow the baseline model's default settings but set 64 hidden cells for all positions. Lastly, we ensure that the training is long enough so that the models converge well; we repeat each of the experiments three times, and report the average results.

### B.3    DIVIDE Architecture

Fig 3 highlights the design of DIVIDE, aligning with Algorithms 1 and 2 introduced in the main paper. Note that the proposed DIVIDE is a hierarchical learning framework, comprising multiple *local* models and a *global*

---

[1] https://numpy.org/
[2] https://scikit-learn.org/stable/
[3] https://pytorch.org/
[4] https://www.pyg.org/
[5] https://github.com/vonum/multi-prophet
[6] https://dtaidistance.readthedocs.io/en/latest/usage/dtw.html
[7] https://pandas.pydata.org/
[8] https://matplotlib.org/

*model.* It benefits from simple collaboration, flexible model selection, and offers efficient training algorithms backed by theoretical guarantees to achieve superior local and global forecasting accuracy.

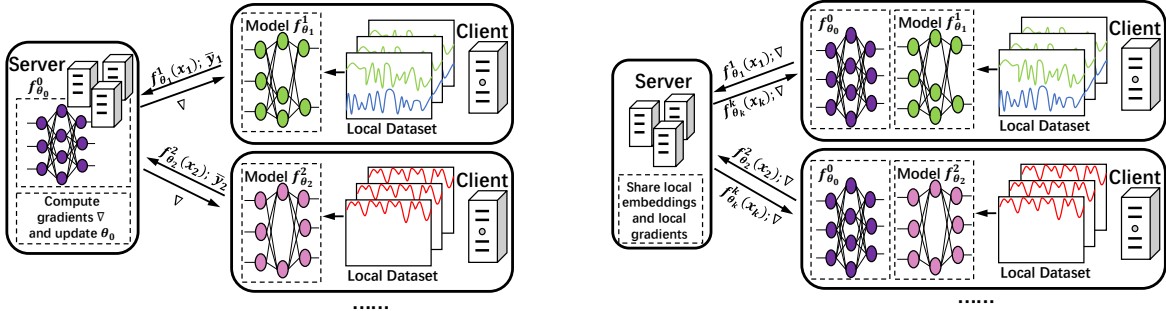

(a) with label sharing.  (b) without label sharing.

Figure 3: Overview of DIVIDE design. Refer to Algorithms 1 and 2 for step-by-step illustrations.

Additionally, we modify the DIVIDE to accommodate the multimodal prediction task, as shown in Fig. 4. In this task, each node adopts a different type of local model based on the modalities of data streaming, reflecting the flexibility of local model selections. Specifically, we employ CNNs for video streams, PointNet for point-cloud data modeling, and RNNs/LSTMs for processing time-series signals. It is also worth noting that our theoretical guarantees still hold.

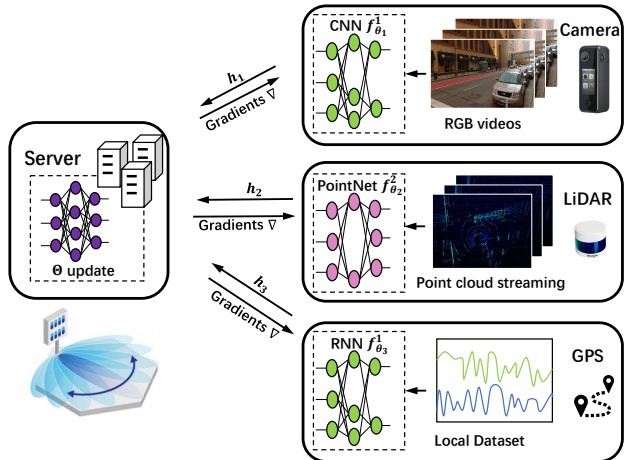

Figure 4: Modified DIVIDE for multimodal prediction task.

## B.4   Training and test strategies

**Neural network training and evaluation strategy.** In line with common time-series forecasting techniques, we employ a sliding window approach for training and assessing the neural networks, as illustrated in Fig. 5b. The model is initially trained on the segmented training dataset and subsequently evaluated. For the divided training and test datasets, data points within the historical window are used to make predictions for the future window. Additionally, we configure the learning epoch to 300, which is relatively generous, considering the model often converges well before the initial epochs. Throughout the training phase, we continuously save the best-performing model on the validation set to ensure convergence for all models.

**Prophet training and evaluation strategies.** Prophet was originally designed for the univariate time-series forecasting task. We utilize the multi-prophet library[9] to extend the Prophet for the multivariate time-series, where each time-series data stream will be used to fit a Prophet independently. We optimize the library via multi-processing programming for speeding up. Note that we use Prophet instead of multi-Prophet for simplicity in the paper.

---

[9]https://github.com/vonum/multi-prophet

We evaluate the prophet model in three schemes, as shown in Fig. 5. We denote them as *Entire*, *EveryIter*, and *Incremental* for simplicity. Specifically, the *Entire* scheme fits the Prophet on the whole training dataset and evaluates the test dataset in one shot. The *EveryIter* only fits the model on the historical data within the sliding window (light blue) and predicts the future (light green). Unlike the neural network, Prophet does not utilize the split training dataset (dark blue). In other words, each sliding window entails the use of a new Prophet model. In contrast, the *Incremental* scheme[10] will continuously update the training set by adding past data samples to it (dark and light blue) as the time-domain sliding window moves forward and refit the Prophet model for prediction, instead of fitting a fixed training set once and predicting the entire future. In this way, the prophet actually has an advantage over the neural network-based model, which is only trained on the fixed training set. However, it should be noted that both *EveryIter* and *Incremental* result in increased computation time compared to neural networks, since they require refitting the model over and over again.

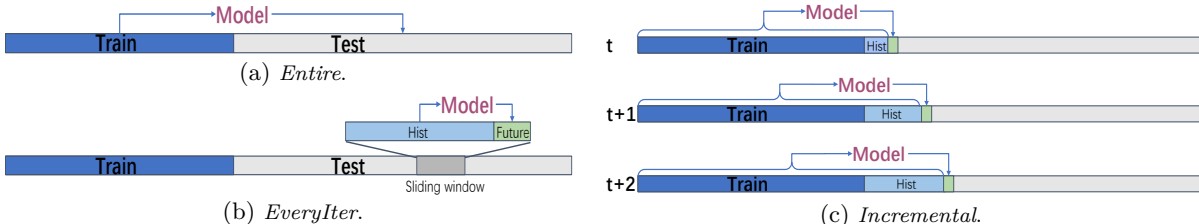

(a) *Entire.*      (b) *EveryIter.*      (c) *Incremental.*

Figure 5: Training and Evaluation Strategies.

## B.5 Evaluation Metrics

**Root Mean Square Error** (RMSE) is a measure of the average magnitude of errors between predicted and observed values:

$$RMSE = \sqrt{\frac{\sum_{i=1}^{N}(x_i - \widehat{x}_i)^2}{N}}$$

where $x_i$ are ground truth observations, and $\widehat{x}_i$ are predicted values of the series. The lower values indicate better predictive performance.

**Dynamic Time Warping** (DTW) is a technique for measuring the similarity between two sequences while accounting for variations in the alignment and pacing of their data points:

$$DTW_q(x, x') = \min_{\pi \in A(x,x')} \left( \sum_{(i,j) \in \pi} d(x_i, x'_j)^q \right)^{\frac{1}{q}}$$

where $\pi$ is an alignment path, $A(x, x')$ is the set of all admissible paths, and $d(x_i, x'_j)$ is a specified distance metric (e.g. Euclidean); see met (2021) for more details. The lower values indicate better predictive performance.

**Distance-Based Accuracy Score** (DBA Score) is a metric for assessing the accuracy of a machine learning model's predictions by averaging the minimum differences between predicted labels and ground truth labels: DBA-Score = $\frac{1}{3}(Y_1 + Y_2 + Y_3)$ where $Y_K$ is defined as:

$$Y_K = 1 - \frac{1}{N} \sum_{n=1}^{N} \min_{1 \leq k \leq K} \min \left( \frac{\|\widehat{y}_{n,k} - y_n\|}{\Delta}, 1 \right)$$

with $y_n$ and $\widehat{y}_{n,k}$ are the ground truth label and the $k$th predicted label respectively. The $k$th predicted label the $k$th most likely label predicted by the machine learning model. $\Delta$ is a normalization factor, in our experiments, we used $\Delta = 5$. The higher values indicate better predictive performance.

---

[10] https://facebook.github.io/prophet/docs/diagnostics.html

# C    Additional numerical results

## C.1    Prophet

Table 4 shows the Prophet's performance when adopting different schemes. We can observe that the *Incremental* scheme achieves the best performance, aligning with the intuitive understanding that it is visible to most data points. This is the scheme we selected and reported in the main paper.

Table 4: Performance of Prophet with different schemes. The lower value indicates better performance.

| Prophet | | POWR | | | | SUBW | | | | WEAT | | | |
|---|---|---|---|---|---|---|---|---|---|---|---|---|---|
| | | 24h | 48h | 72h | 168h | 24h | 48h | 72h | 168h | 24h | 48h | 72h | 168h |
| Entire | RMSE | 0.0858 | 0.0918 | 0.0949 | 0.1020 | 0.1517 | 0.1542 | 0.1734 | 0.1856 | 0.0263 | 0.0331 | 0.0398 | 0.0399 |
| | DTW | 0.3621 | 0.4956 | 0.5564 | 0.8633 | 0.5382 | 0.7407 | 0.9251 | 1.4120 | 0.0861 | 0.1398 | 0.1458 | 0.2545 |
| EveryIter | RMSE | 0.0765 | 0.0834 | 0.0879 | 0.0891 | 0.1322 | 0.1398 | 0.1523 | 0.1479 | 0.0249 | 0.0289 | 0.0354 | 0.0358 |
| | DTW | 0.3339 | 0.4487 | 0.5060 | 0.7587 | 0.4982 | 0.7043 | 0.8083 | 1.2661 | 0.0803 | 0.1205 | 0.1318 | 0.2285 |
| Incr. | RMSE | **0.0764** | **0.0787** | **0.0791** | **0.0820** | **0.1242** | **0.1361** | **0.1440** | **0.1452** | **0.0204** | **0.0211** | **0.0215** | **0.0246** |
| | DTW | **0.3117** | **0.4315** | **0.4850** | **0.7017** | **0.4717** | **0.6832** | **0.7575** | **1.1922** | **0.0801** | **0.1179** | **0.1314** | **0.2210** |

## C.2    Multivariate time-series forecasting

In this subsection, we provide additional numerical results on multivariate time-series forecasting, covering hierarchical tasks, mixed local models, the global model's impact, and asynchronous updates.

**Hierarchical tasks:** We now further introduce two global tasks referred to as *Sum* and *Spatial* for simplicity. In the *Sum* task, the global prediction aims to capture data streams that possess an algebraic correlation with all local nodes. For instance, it may involve predicting a country's power demand as the sum of demands from each region. To address this, we modify the baseline methods and present the results in Table 5. In the case of Prophet and DIVIDE_NoG, where global models are absent and individual data streams are considered separately, we calculate the global prediction based on the aggregation of all local results. In the centralized model, an additional output dimension is introduced. For DIVIDE_MLP, we employ the global prediction model.

Table 5: The study of *Sum* task. We examine the DIVIDE without a global model (DIVIDE_NoG) and with MLP as the global model (DIVIDE_MLP), as well as Prophet and a centralized LSTM model. Each model is supposed to utilize historical 36-hrs data to predict future {24,48,72,168}-hrs.

| Global Task | Model Metrics | Prophet | | DIVIDE(LSTM)_NoG | | LSTM | | DIVIDE(LSTM)_MLP | |
|---|---|---|---|---|---|---|---|---|---|
| | | RMSE | DTW | RMSE | DTW | RMSE | DTW | RMSE | DTW |
| Sum / POWR | 24h | 0.094 | 0.109 | 0.071 | 0.093 | 0.067 | **0.090** | **0.060** | 0.095 |
| | 48h | 0.131 | 0.252 | 0.079 | 0.196 | 0.076 | 0.171 | **0.070** | **0.152** |
| | 72h | 0.167 | 0.426 | 0.120 | 0.236 | 0.108 | 0.192 | **0.106** | **0.178** |
| | 168h | 0.191 | 0.653 | 0.142 | 0.402 | 0.147 | 0.381 | **0.132** | **0.354** |
| Methodology | | Sum | | Sum | | One output dim in LSTM | | Global model output | |
| Sharing info | | Local node predictions | | Local node predictions | | Raw data | | Local node hidden states | |

Meanwhile, we also consider the *Spatial* task, which is supposed to predict one of the local node information based on the others. For example, one region may lose monitoring, and we want to use the others to predict the missing value. We report the results in Table 6. In this task, both the Prophet and DIVIDE_NoG have no way to do it due to the lack of a global model. From those results, the hierarchical design of DIVIDE demonstrates its adaptability to various tasks and exhibits superior performance in global tasks.

**Scalability:** We vary the number of nodes in the system and study the impact of the global model on the scalability. Fig. 6 reports the results of applying DIVEIDE(LSTM) on the POWR dataset with and without the global model. We can observe that (1) the DIVIDE with MLP consistently performs better than the no global model's version; (2) having MLP as the global model makes the performance insensitive to the number of nodes, while not having it leads to significantly worse results as the number of nodes increases. These observations are consistent when applying the other local model on different datasets.

Table 6: The study of *Spatial* task.

| Global Task | Model Metrics | | Prophet RMSE | DTW | DIVIDE(LSTM)_NoG RMSE | DTW | LSTM RMSE | DTW | DIVIDE(LSTM)_MLP RMSE | DTW |
|---|---|---|---|---|---|---|---|---|---|---|
| Spatial | POWR | 24h | X | X | X | X | 0.107 | 0.180 | **0.096** | **0.161** |
| | | 48h | X | X | X | X | 0.186 | 0.208 | **0.172** | **0.206** |
| | | 72h | X | X | X | X | 0.207 | 0.497 | **0.203** | **0.442** |
| | | 168h | X | X | X | X | **0.274** | 0.669 | 0.297 | **0.583** |
| Methodology | | | No way to do so | | No way to do so | | One output dim in LSTM | | Global model output | |
| Sharing info | | | X | | X | | Raw data | | Local node hidden states | |

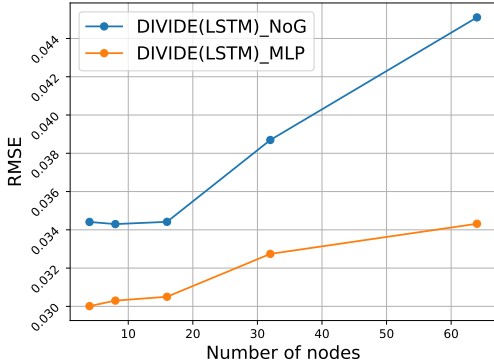

Figure 6: Impact of the global model on the framework scalability. The experiments are conducted on the POWR dataset with {4,8,16,32,64} nodes running the LSTM on the local nodes. The model is supposed to utilize a historical 36-hour window to predict the future 24 hours.

**Asynchronous model updates:** We conduct experiments with different learning rates to the local and global updates to study the impact of asynchronous model updates. The results are presented in Table 7. The results demonstrate that (1) our framework can flexibly operate on various configurations of global and local learning rates, and (2) it performs robustly across different learning rate choices, verifying the statement presented in the main paper above.

Table 7: The performance of DIVIDE(LSTM) on the POWR dataset with different global and local learning rates using RMSE. The model is supposed to utilize a historical 36hrs window to predict future 24hrs. The $\eta_0$ in $\textbf{global}_{\eta_0}$ represents the learning rate of the global model while $\eta_k$ in $\textbf{local}_{\eta_k}$ denotes the local learning rate of the client nodes. The bold configuration is the one we used in the rest of the paper by default.

| Learning Rate | $\textbf{Local}_{0.001}$ | $\textbf{Local}_{0.003}$ | $\textbf{Local}_{0.01}$ | $\textbf{Local}_{0.03}$ | $\textbf{Local}_{0.1}$ |
|---|---|---|---|---|---|
| $\textbf{Global}_{0.001}$ | 0.0330 | 0.0339 | 0.0298 | 0.0318 | 0.0345 |
| $\textbf{Global}_{0.003}$ | 0.0288 | 0.0275 | 0.0293 | 0.0342 | 0.0338 |
| $\textbf{Global}_{0.01}$ | 0.0295 | 0.0294 | **0.0303** | 0.0307 | 0.0291 |
| $\textbf{Global}_{0.03}$ | 0.0315 | 0.0298 | 0.0309 | 0.0306 | 0.0326 |
| $\textbf{Global}_{0.1}$ | 0.0301 | 0.0290 | 0.0324 | 0.0312 | 0.0308 |

### C.3 Multimodal prediction

We follow the ITU Challenge setups (ITU, 2022) for the multimodal prediction task. In this setup, the training dataset consists of 80% of the data from scenarios 32∼34, and evaluation is conducted on the test dataset, which includes scenario 31 and previously unseen data from scenarios 32∼34. The DBA score is employed as the performance metric, and the overall DBA-Score is calculated by averaging the results across these four scenarios, serving as the metric for the final ranking. Table 8 reports the numerical results achieved

by the DIVIDE. It is worth noting that DIVIDE ranks in the top 5 among the reported $150+$ submissions[11], proving evidence of its capability to efficiently fuse and utilize different data modalities.

Meanwhile, we observe that DIVIDE performs worst in scenario 31 and speculate that this could be attributed to a lack of data pre-processing. For example, the brightness has a significant impact on the video, and thus, the superior solutions adopt comprehensive data augmentation methods to adjust and normalize the raw datasets across all the scenarios, but it is beyond this paper's scope as we emphasize the ability of modality fusion and flexibility of local model selection instead of simply beating the best solution in the competition via data pre-processing. It's also worth noting that our model hasn't undergone fine-tuning using the adaptation dataset provided by ITU, which is another significant factor contributing to the poor performance in scenario 31.

Table 8: DIVIDE performance on multimodal prediction tasks evaluated on scenarios 31-34 of the DS6G datasets. The DBA score evaluates the performance, with the higher number indicating better performance. The last row reports the ranks based on the public leaderboard and is determined based on when all modalities are utilized.

|  | Scenario 31 | Scenario 32 | Scenario 33 | Scenario 34 | Overall |
|---|---|---|---|---|---|
| **GPS-only** | 0.0256 | 0.6518 | 0.6332 | 0.6264 | 0.4843 |
| **Vision-only** | 0.1142 | 0.6163 | 0.6830 | 0.6619 | 0.5189 |
| **LiDAR-only** | 0.0653 | 0.6385 | 0.6120 | 0.5967 | 0.4781 |
| **DIVIDE(All)** | 0.0715 | 0.7450 | 0.6895 | 0.7385 | 0.5611 |
| **Rank** | 13 | 2 | 10 | 6 | 5 |

# D    Proof of Theorem 4.1

Here, we restate Theorem 4.1 for convenience.

**Theorem D.1.** *Suppose Assumptions 2 and 3 hold, and that the learning rates satisfy:*

$$\sum_r \eta_k^r = +\infty, \ \ \sum_r \ln^2 r \cdot (\eta_k^r)^2 < +\infty \ \ \forall k \in \{0, 1, \ldots, K\}.$$

*Then we have* $\lim_{R \to \infty} \mathbb{E}\|\nabla_\theta \mathcal{L}(\theta)\| = 0$. *Moreover, we have*

$$\mathbb{E}\Big[ \min_{1 \le r \le R} \{\|\nabla_\theta \mathcal{L}(\theta^r)\|^2\} \Big] = \mathcal{O}\Bigg( \frac{\Psi(T)}{\sum_{r=1}^R \min\{\eta_k^r\}_{k=0}^K} \Bigg),$$

*where* $\theta^r := [(\theta_0^r)^T, \ldots, (\theta_K^r)^T]^T$ *and* $\Psi(T)$ *is*

$$\Psi(T) := \max\Big\{1, \frac{1}{\ln(1/\lambda(T))}\Big\}, \ \ with \ \lambda(T) := \frac{\max\{|\lambda_2(T)|, |\lambda_M(T)|\} + 1}{2} \in [0, 1),$$

*where* $\lambda_i(T) \in \mathbb{C}$ *is the* $i^{th}$ *largest eigenvalue of* $T$.

To prove the Theorem D.1 we will utilize some intermediate results. First, let us set up some notations. Recall, that our goal is to solve the following problem

$$\min_{\theta \in \Theta \subseteq \mathbb{R}^P} \Big\{ \mathcal{L}(\theta) := \mathbb{E}_{(\bar{\mathbf{x}}, \bar{\mathbf{y}}) \sim \Pi}[\mathcal{L}(f_0(\theta_0; f_1(\theta_1), \ldots, f_K(\theta_K)); (\bar{\mathbf{x}}, \bar{\mathbf{y}}))] \Big\}, \tag{8}$$

Let us consider the (Markov chain) SGD-based update rule to solve (8) in Algorithms 1 and 2. Note the overall (network-wide) stochastic gradient computed at round $r \in \{0, 1, \ldots, R-1\}$ is

$$\nabla \mathcal{L}(\theta^r; (\bar{\mathbf{x}}^r, \bar{\mathbf{y}}^r)) = [\nabla_{\theta_0} \mathcal{L}(\theta^r; (\bar{\mathbf{x}}^r, \bar{\mathbf{y}}^r))^T, \nabla_{\theta_1} \mathcal{L}(\theta^r; (\bar{\mathbf{x}}^r, \bar{\mathbf{y}}^r))^T, \ldots, \nabla_{\theta_K} \mathcal{L}(\theta^r; (\bar{\mathbf{x}}^r, \bar{\mathbf{y}}^r))^T]^T.$$

---

[11]https://www.deepsense6g.net/ml-task-multi-modal-beam-prediction/

Moreover, the Markov chain gradient descent update rule at each node and the server is

$$\theta_i^{r+1} = \theta_i^r - \eta_i^r \nabla_{\theta_i^r} \mathcal{L}(\theta^r; (\bar{\mathbf{x}}^r, \bar{\mathbf{y}}^r)) \quad \text{for} \quad i = \{0, 1, \ldots, K\} \tag{9}$$

Next, we rewrite problem (8) in a finite-sum version. Let us suppose that the distribution $\Pi$ is supported on a set of $M$ points $\xi^1, \ldots, \xi^M$, then problem (8) above can be stated equivalently as a finite-sum problem

$$\min_{\theta \in \Theta \subseteq \mathbb{R}^P} \left\{ \mathcal{L}(\theta) := \frac{1}{M} \sum_{m=1}^{M} \mathcal{L}_m(\theta) \right\}, \tag{10}$$

where $\mathcal{L}_m(\cdot)$ is defined as

$$\mathcal{L}_m(\theta) := M \cdot \mathbb{P}[(\bar{\mathbf{x}}, \bar{\mathbf{y}}) = \xi^m] \cdot \mathcal{L}(f_0(\theta_0; f_1(\theta_1), \ldots, f_K(\theta_K)); \xi^m)$$

Note here that each state $m \in [M]$ has uniform probability $1/M$. The corresponding version of the Markov chain gradient descent for problem (10) can be written as

$$\theta_i^{r+1} = \theta_i^r - \eta_i^r \nabla_{\theta_i^r} \mathcal{L}_{m^r}(\theta^r) \quad \text{for} \quad i = \{0, 1, \ldots, K\} \tag{11}$$

where $\{m^r\}_{r \geq 0}$ is the trajectory of a Markov chain on $\{1, 2, \ldots, M\}$ that has uniform stationary distribution. Note that the Markov chain trajectories $\{(\bar{\mathbf{x}}^r, \bar{\mathbf{y}}^r)\}_{r \geq 0}$ and $\{m^r\}_{r \geq 0}$ are two different but related Markov chains. A crucial difference in the update rules of (11) and (9) from a centralized setting is that the local updates at each node here are heterogeneous with potentially different learning rates (Sun et al., 2018). Next, we make the following assumption on the underlying Markov chain generating the trajectory $\{m^r\}_{r \geq 0}$.

**Assumption 4.** We assume that the data-generating process $\{m^r\}_{r \geq 0}$ follows a Markov chain trajectory with $M$ states. The Markov chain is time-homogeneous, irreducible, and aperiodic. The Markov chain has a transition matrix $T \in \mathbb{R}^{M \times M}$ and stationary distribution $\pi^* := [\pi_1^*, \ldots, \pi_M^*]$ with $\sum_{m=1}^{M} \pi_m^* = 1$.

Moreover, for analysis purposes, we make the following assumptions.

**Assumption 5.** The **Local SG** and the **Global SG** in (11) are bounded, i.e., we have $\|\nabla_{\theta_k} \mathcal{L}_m(\theta)\| \leq G$ and $\|\nabla_{\theta_0} \mathcal{L}_m(\theta)\| \leq G$ for each $m \in [M]$. We also assume that the loss function is $L$-Lipschitz smooth, i.e., $\nabla_\theta \mathcal{L}_m(\theta)$ is $L$-Lipschitz.

We note that for appropriately chosen $L$ and $G$ Assumption 5 above and Assumption 3 in the main paper are equivalent. We define

$$\mathcal{T}_r := \min \left\{ \max \left\{ \left\lceil \ln \left( \frac{r}{2C_T G^2 K^2} \middle/ \ln \left( \frac{1}{\lambda(T)} \right) \right) \right\rceil, K_T \right\}, r \right\}$$

where $C_T$ is a constant that depends on the Jordan canonical form of $T$ and $K_T$ is a constant that depends on $\lambda(T)$ and $\lambda_2(T)$. Please see (Sun et al., 2018, Lemma 1). This choice of $\mathcal{T}_r$ implies that we have from (Sun et al., 2018, Lemma 1)

$$\left| [T^{\mathcal{T}_r}]_{i,j} - \frac{1}{M} \right| \leq \frac{1/r}{2G^2 K^2} \quad \text{for any} \quad i, j \in \{1, 2, \ldots, M\}.$$

Specifically, $\mathcal{T}_r$ characterizes the mixing time of the Markov chain. Next, we define by $\mathcal{F}^r$ the sigma-algebra generated by the sequence of iterates $\theta^r$ as

$$\mathcal{F}^r := \sigma(\theta^1, \theta^2, \ldots, \theta^r, m^0, m^1, \ldots, m^{r-1}).$$

where $m^r$ are generated according to Assumption 4.

*Proof.* Using the Lipschitz smoothness of $\mathcal{L}(\cdot)$, we have

$$
\begin{aligned}
\mathcal{L}(\theta^{r+1}) - \mathcal{L}(\theta^r) &\leq \langle \nabla \mathcal{L}(\theta^r), \theta^{r+1} - \theta^r \rangle + \frac{L}{2} \|\theta^{r+1} - \theta^r\|^2 \\
&= \langle \nabla \mathcal{L}(\theta^{r-\mathcal{T}_r}), \theta^{r+1} - \theta^r \rangle + \langle \nabla \mathcal{L}(\theta^r) - \mathcal{L}(\theta^{r-\mathcal{T}_r}), \theta^{r+1} - \theta^r \rangle + \frac{L}{2} \|\theta^{r+1} - \theta^r\|^2 \\
&\leq \langle \nabla \mathcal{L}(\theta^{r-\mathcal{T}_r}), \theta^{r+1} - \theta^r \rangle + \frac{1}{2} \|\nabla \mathcal{L}(\theta^r) - \mathcal{L}(\theta^{r-\mathcal{T}_r})\|^2 + \frac{L+1}{2} \|\theta^{r+1} - \theta^r\|^2 \\
&\leq \langle \nabla \mathcal{L}(\theta^{r-\mathcal{T}_r}), \theta^{r+1} - \theta^r \rangle + \frac{L^2}{2} \|\theta^r - \theta^{r-\mathcal{T}_r}\|^2 + \frac{L+1}{2} \|\theta^{r+1} - \theta^r\|^2
\end{aligned}
$$

where the first inequality follows from the Schwarz inequality and the final inequality results from the Lipschitz-smoothness of $\mathcal{L}(\cdot)$. Rearranging the terms, we get

$$
\langle \nabla \mathcal{L}(\theta^{r-\mathcal{T}_r}), -(\theta^{r+1} - \theta^r) \rangle \leq \mathcal{L}(\theta^r) - \mathcal{L}(\theta^{r+1}) + \frac{L^2}{2} \|\theta^r - \theta^{r-\mathcal{T}_r}\|^2 + \frac{L+1}{2} \|\theta^{r+1} - \theta^r\|^2
$$

Next, we consider the l.h.s. of the above equation. Taking expectation w.r.t. the sigma-algebra $\mathcal{F}^{r-\mathcal{T}_r}$, we get

$$
\mathbb{E}[\langle \nabla \mathcal{L}(\theta^{r-\mathcal{T}_r}), -(\theta^{r+1} - \theta^r)\rangle | \mathcal{F}^{r-\mathcal{T}_r}] = \mathbb{E}\left[\left\langle \nabla \mathcal{L}(\theta^{r-\mathcal{T}_r}), \begin{bmatrix} \eta_0^r \nabla_{\theta_0} \mathcal{L}_{m^r}(\theta^r) \\ \eta_1^r \nabla_{\theta_1} \mathcal{L}_{m^r}(\theta^r) \\ \vdots \\ \eta_K^r \nabla_{\theta_K} \mathcal{L}_{m^r}(\theta^r) \end{bmatrix} \right\rangle \Big| \mathcal{F}^{r-\mathcal{T}_r}\right]
$$

$$
= \mathbb{E}\left[\left\langle \nabla \mathcal{L}(\theta^{r-\mathcal{T}_r}), \begin{bmatrix} \eta_0^r \nabla_{\theta_0} \mathcal{L}_{m^r}(\theta^{r-\mathcal{T}_r}) \\ \eta_1^r \nabla_{\theta_1} \mathcal{L}_{m^r}(\theta^{r-\mathcal{T}_r}) \\ \vdots \\ \eta_K^r \nabla_{\theta_K} \mathcal{L}_{m^r}(\theta^{r-\mathcal{T}_r}) \end{bmatrix} \right\rangle \Big| \mathcal{F}^{r-\mathcal{T}_r}\right]
$$

$$
+ \mathbb{E}\left[\left\langle \nabla \mathcal{L}(\theta^{r-\mathcal{T}_r}), \begin{bmatrix} \eta_0^r \nabla_{\theta_0} \mathcal{L}_{m^r}(\theta^r) \\ \eta_1^r \nabla_{\theta_1} \mathcal{L}_{m^r}(\theta^r) \\ \vdots \\ \eta_K^r \nabla_{\theta_K} \mathcal{L}_{m^r}(\theta^r) \end{bmatrix} - \begin{bmatrix} \eta_0^r \nabla_{\theta_0} \mathcal{L}_{m^r}(\theta^{r-\mathcal{T}_r}) \\ \eta_1^r \nabla_{\theta_1} \mathcal{L}_{m^r}(\theta^{r-\mathcal{T}_r}) \\ \vdots \\ \eta_K^r \nabla_{\theta_K} \mathcal{L}_{m^r}(\theta^{r-\mathcal{T}_r}) \end{bmatrix} \right\rangle \Big| \mathcal{F}^{r-\mathcal{T}_r}\right]
$$

$$
\geq \mathbb{E}\left[\left\langle \nabla \mathcal{L}(\theta^{r-\mathcal{T}_r}), \begin{bmatrix} \eta_0^r \nabla_{\theta_0} \mathcal{L}_{m^r}(\theta^{r-\mathcal{T}_r}) \\ \eta_1^r \nabla_{\theta_1} \mathcal{L}_{m^r}(\theta^{r-\mathcal{T}_r}) \\ \vdots \\ \eta_K^r \nabla_{\theta_K} \mathcal{L}_{m^r}(\theta^{r-\mathcal{T}_r}) \end{bmatrix} \right\rangle \Big| \mathcal{F}^{r-\mathcal{T}_r}\right]
$$

$$
- K \cdot G \cdot L \sum_{i=0}^{K} \eta_i^r \mathbb{E}\left[\|\theta^r - \theta^{r-\mathcal{T}_r}\| \Big| \mathcal{F}^{r-\mathcal{T}_r}\right]
$$

Therefore, we get

$$
\mathbb{E}\left[\left\langle \nabla \mathcal{L}(\theta^{r-\mathcal{T}_r}), \begin{bmatrix} \eta_0^r \nabla_{\theta_0} \mathcal{L}_{m^r}(\theta^{r-\mathcal{T}_r}) \\ \eta_1^r \nabla_{\theta_1} \mathcal{L}_{m^r}(\theta^{r-\mathcal{T}_r}) \\ \vdots \\ \eta_K^r \nabla_{\theta_K} \mathcal{L}_{m^r}(\theta^{r-\mathcal{T}_r}) \end{bmatrix} \right\rangle \Big| \mathcal{F}^{r-\mathcal{T}_r}\right] \leq \mathbb{E}\left[\mathcal{L}(\theta^r) - \mathcal{L}(\theta^{r+1}) \big| \mathcal{F}^{r-\mathcal{T}_r}\right]
$$

$$
+ \frac{L^2}{2} \mathbb{E}\left[\|\theta^r - \theta^{r-\mathcal{T}_r}\|^2 \big| \mathcal{F}^{r-\mathcal{T}_r}\right] + \frac{L+1}{2} \mathbb{E}\left[\|\theta^{r+1} - \theta^r\|^2 \big| \mathcal{F}^{r-\mathcal{T}_r}\right]
$$

$$
+ K \cdot G \cdot L \sum_{i=0}^{K} \eta_i^r \mathbb{E}\left[\|\theta^r - \theta^{r-\mathcal{T}_r}\| \Big| \mathcal{F}^{r-\mathcal{T}_r}\right].
$$

Taking expectations on both sides, we get

$$\mathbb{E}\left[\left\langle \nabla\mathcal{L}(\theta^{r-\mathcal{T}_r}), \begin{bmatrix} \eta_0^r \nabla_{\theta_0}\mathcal{L}_{m^r}(\theta^{r-\mathcal{T}_r}) \\ \eta_1^r \nabla_{\theta_1}\mathcal{L}_{m^r}(\theta^{r-\mathcal{T}_r}) \\ \vdots \\ \eta_K^r \nabla_{\theta_K}\mathcal{L}_{m^r}(\theta^{r-\mathcal{T}_r}) \end{bmatrix} \right\rangle\right] \le \mathbb{E}\left[\mathcal{L}(\theta^r) - \mathcal{L}(\theta^{r+1})\right] + \frac{L+1}{2}\mathbb{E}\left[\|\theta^{r+1} - \theta^r\|^2\right]$$

$$+ K \cdot G \cdot L \sum_{i=0}^K \eta_i^r \mathbb{E}\left[\|\theta^r - \theta^{r-\mathcal{T}_r}\|\right] + \frac{L^2}{2}\mathbb{E}\left[\|\theta^r - \theta^{r-\mathcal{T}_r}\|^2\right].$$

Next, we consider the l.h.s.term

$$\mathbb{E}_{m^r}\left[\left\langle \nabla\mathcal{L}(\theta^{r-\mathcal{T}_r}), \begin{bmatrix} \eta_0^r \nabla_{\theta_0}\mathcal{L}_{m^r}(\theta^{r-\mathcal{T}_r}) \\ \eta_1^r \nabla_{\theta_1}\mathcal{L}_{m^r}(\theta^{r-\mathcal{T}_r}) \\ \vdots \\ \eta_K^r \nabla_{\theta_K}\mathcal{L}_{m^r}(\theta^{r-\mathcal{T}_r}) \end{bmatrix} \right\rangle \bigg| \mathcal{F}^{r-\mathcal{T}_r}\right]$$

$$= \sum_{m=1}^M \left\langle \nabla\mathcal{L}(\theta^{r-\mathcal{T}_r}), \begin{bmatrix} \eta_0^r \nabla_{\theta_0}\mathcal{L}_{m^r}(\theta^{r-\mathcal{T}_r}) \\ \eta_1^r \nabla_{\theta_1}\mathcal{L}_{m^r}(\theta^{r-\mathcal{T}_r}) \\ \vdots \\ \eta_K^r \nabla_{\theta_K}\mathcal{L}_{m^r}(\theta^{r-\mathcal{T}_r}) \end{bmatrix} \right\rangle \cdot \mathbb{P}[m^r = m | \mathcal{F}^{r-\mathcal{T}_r}]$$

$$= \sum_{m=1}^M \left\langle \nabla\mathcal{L}(\theta^{r-\mathcal{T}_r}), \begin{bmatrix} \eta_0^r \nabla_{\theta_0}\mathcal{L}_{m^r}(\theta^{r-\mathcal{T}_r}) \\ \eta_1^r \nabla_{\theta_1}\mathcal{L}_{m^r}(\theta^{r-\mathcal{T}_r}) \\ \vdots \\ \eta_K^r \nabla_{\theta_K}\mathcal{L}_{m^r}(\theta^{r-\mathcal{T}_r}) \end{bmatrix} \right\rangle \cdot \mathbb{P}[m^r = m | m^{r-\mathcal{T}_r}]$$

$$= \sum_{m=1}^M \left\langle \nabla\mathcal{L}(\theta^{r-\mathcal{T}_r}), \begin{bmatrix} \eta_0^r \nabla_{\theta_0}\mathcal{L}_{m^r}(\theta^{r-\mathcal{T}_r}) \\ \eta_1^r \nabla_{\theta_1}\mathcal{L}_{m^r}(\theta^{r-\mathcal{T}_r}) \\ \vdots \\ \eta_K^r \nabla_{\theta_K}\mathcal{L}_{m^r}(\theta^{r-\mathcal{T}_r}) \end{bmatrix} \right\rangle \cdot [T^{\mathcal{T}_r}]_{m^r-\mathcal{T}_r, m}$$

$$= \left\langle \nabla\mathcal{L}(\theta^{r-\mathcal{T}_r}), \sum_{m=1}^M [T^{\mathcal{T}_r}]_{m^r-\mathcal{T}_r, m} \begin{bmatrix} \eta_0^r \nabla_{\theta_0}\mathcal{L}_{m^r}(\theta^{r-\mathcal{T}_r}) \\ \eta_1^r \nabla_{\theta_1}\mathcal{L}_{m^r}(\theta^{r-\mathcal{T}_r}) \\ \vdots \\ \eta_K^r \nabla_{\theta_K}\mathcal{L}_{m^r}(\theta^{r-\mathcal{T}_r}) \end{bmatrix} \right\rangle.$$

$$\ge \min\{\eta_i^r\}_{i=0}^K \|\nabla\mathcal{L}(\theta^{r-\mathcal{T}_r})\|^2 - \frac{\max\{\eta_i^r\}_{i=0}^K}{2r}$$

Therefore, we finally get

$$\min\{\eta_i^r\}_{i=0}^K \mathbb{E}\left[\|\nabla\mathcal{L}(\theta^{r-\mathcal{T}_r})\|^2\right] \le \mathbb{E}\left[\mathcal{L}(\theta^r) - \mathcal{L}(\theta^{r+1})\right] + \frac{L+1}{2}\mathbb{E}\left[\|\theta^{r+1} - \theta^r\|^2\right]$$

$$+ K \cdot G \cdot L \sum_{i=0}^K \eta_i^r \mathbb{E}\left[\|\theta^r - \theta^{r-\mathcal{T}_r}\|\right] + \frac{L^2}{2}\mathbb{E}\left[\|\theta^r - \theta^{r-\mathcal{T}_r}\|^2\right] + \frac{\max\{\eta_i^r\}_{i=0}^K}{2r}$$

Summing over $r$ and using (Sun et al., 2018, Equations (6.58),(6.59), and (6.60)), we get the following

$$\sum_r \min\{\eta_i^r\}_{i=0}^K \mathbb{E}\left[\|\nabla\mathcal{L}(\theta^{r-\mathcal{T}_r})\|^2\right] \le \mathcal{O}\left(\max\left\{1, \frac{1}{\ln(1/\lambda(T))}\right\}\right) \tag{12}$$

Next, using the Lipschitz-smoothness of $\mathcal{L}(\cdot)$, we have

$$\|\nabla\mathcal{L}(\theta^r)\|^2 - \|\nabla\mathcal{L}(\theta^{r-\mathcal{T}_r})\|^2 \le \left\langle \nabla\mathcal{L}(\theta^r) - \nabla\mathcal{L}(\theta^{r-\mathcal{T}_r}), \nabla\mathcal{L}(\theta^r) + \nabla\mathcal{L}(\theta^{r-\mathcal{T}_r})\right\rangle$$

$$\le \|\nabla\mathcal{L}(\theta^r) - \nabla\mathcal{L}(\theta^{r-\mathcal{T}_r})\| \cdot \|\nabla\mathcal{L}(\theta^r) + \nabla\mathcal{L}(\theta^{r-\mathcal{T}_r})\|$$

$$\le 2K \cdot G \cdot L \cdot \|\theta^r - \theta^{r-\mathcal{T}_r}\|$$

Multiplying both sides by $\min\{\eta_i^r\}_{i=0}^K$, we get

$$
\begin{aligned}
\min\{\eta_i^r\}_{i=0}^K &\cdot \|\nabla\mathcal{L}(\theta^r)\|^2 - \min\{\eta_i^r\}_{i=0}^K \cdot \|\nabla\mathcal{L}(\theta^{r-\mathcal{T}_r})\|^2 \\
&\leq 2K \cdot G \cdot L \cdot \min\{\eta_i^r\}_{i=0}^K \cdot \|\theta^r - \theta^{r-\mathcal{T}_r}\| \\
&\leq K \cdot G \cdot L \cdot \min\{\eta_i^r\}_{i=0}^K \cdot \|\theta^r - \theta^{r-\mathcal{T}_r}\|^2 + K \cdot G \cdot L \cdot \left(\min\{\eta_i^r\}_{i=0}^K\right)^2
\end{aligned}
$$

Again summing over $r$, taking expectation, and using the discussion in (Sun et al., 2018, Equations (6.58),(6.59), and (6.60)), we get the

$$
\sum_r \left(\min\{\eta_i^r\}_{i=0}^K \cdot \mathbb{E}\|\nabla\mathcal{L}(\theta^r)\|^2 - \min\{\eta_i^r\}_{i=0}^K \cdot \mathbb{E}\|\nabla\mathcal{L}(\theta^{r-\mathcal{T}_r})\|^2\right) \leq C_1 + \frac{C_2}{\ln(1/\lambda(T))} \tag{13}
$$

for some constants $C_1, C_2 > 0$. Finally, summing (12) and (13), we get the

$$
\sum_r \min\{\eta_i^r\}_{i=0}^K \cdot \mathbb{E}\|\nabla\mathcal{L}(\theta^r)\|^2 \leq \mathcal{O}\left(\max\left\{1, \frac{1}{\ln(1/\lambda(T))}\right\}\right),
$$

Therefore, we have the proof of the main statement of the Theorem. Moreover, the proof of $\lim_{R\to\infty} \mathbb{E}\|\nabla_\theta\mathcal{L}(\theta)\| = 0$ is a straightforward extension of the proof from (Sun et al., 2018, Theorem 2, Equation (4.4)).

Hence, the theorem is proved. $\qquad\square$

