# OpenReview forum: "Distributed Hierarchical Decomposition Framework for Multimodal Timeseries Prediction"
_TMLR — Accepted by TMLR_

### Review · Reviewer_mogC · 2025-07-28

**Summary Of Contributions:**

### Summary Of Contributions
This paper is concerned with how the hierachical learning which incorporate global and local model influences the distributed time series forecasting task. The authors propose DIVIDE and theoretical analysis demonstrating the framework's effectiveness and justifying the guarantees of convergence.
### Strength And Weaknesses
Strength:
1. The paper is well motivated.
2. The authors show theoretical and empirical contribution of using DIVIDE framework, which harness both global and local models in distributed settings.
3. The results of ITU 2022 challenge show good performance on heterogeneous dataset.

Weaknesses:
1. The authors propose to use global models to capture the correlations among different local nodes. It seems to me that the idea is similar using a virtual latent master node which are present in [1]. I would like to ask the authors if they can relate to the proposed algorithm with it.
2. The proposed mechanism where local nodes shares its embeddings $f_k(\theta_k^r) \ \forall k$ with all nodes appears structurally similar to a fully connected GNN. It would be valuable if the authors could clarify how their approach differs from or improves upon fully connected GNN architectures, particularly in terms of the information flow patterns and computational mechanism.

[1] Neural Message Passing for Quantum Chemistry

**Audience:**

Yes

**Audience Explanation:**

The paper addresses the practically important problem of distributed time series forecasting, which is highly relevant to TMLR audience working on federated learning, and time series analysis applications.

**Claims And Evidence:**

No

**Claims Explanation:**

While the authors present numerical results across multiple benchmark datasets, The distributed DIVIDE framework rarely demonstrates superior performance compared to centralized models across several datasets.

**Requested Changes:**

Please see the questions in the weaknesses section in the review.

---

> ### Author Response · Authors · 2025-09-09
> **Response to reviewer mogC**
>
> **(1) The authors propose to use global models to capture the correlations among different local nodes.** It seems to me that the idea is similar using a virtual latent master node which are present in [1]. I would like to ask the authors if they can relate to the proposed algorithm with it.
> [1] Neural Message Passing for Quantum Chemistry
> **Response:** We thank the reviewer for noting the connection to the latent “master” node in [1]. Both approaches use a global coordination mechanism to capture cross-node dependencies. The key differences are that [1] embeds the master node in a fully centralized graph model for molecular property prediction, whereas our framework is distributed, designed for multivariate time series forecasting under privacy and communication constraints, aggregating and disseminating global information without exposing raw data. We will revise the manuscript to clarify this relationship.
>
>
>
>
>
>
>
> **(2) The proposed mechanism where local nodes shares its embeddings with all nodes appears structurally similar to a fully connected GNN.** It would be valuable if the authors could clarify how their approach differs from or improves upon fully connected GNN architectures, particularly in terms of the information flow patterns and computational mechanism.
> **Response:** We thank the reviewer for this observation. While the information exchange pattern in our framework may appear similar to a fully connected GNN, there are key differences. In a fully connected GNN, message passing occurs directly between all node pairs within a centralized model, requiring full access to the entire graph and incurring quadratic communication cost. In contrast, our approach operates in a distributed setting: local nodes share embeddings only with a global model, which aggregates and redistributes information without direct peer-to-peer (local node to local node) exchange. This design reduces communication overhead, preserves data privacy by avoiding raw data sharing, and supports scalable optimization across distributed environments. We will revise the manuscript to explicitly highlight these differences in information flow and computation.
>
>
>
> **(3) While the authors present numerical results across multiple benchmark datasets, The distributed DIVIDE framework rarely demonstrates superior performance compared to centralized models across several datasets.**
> **Response:** We thank the reviewer for raising this concern. We emphasize that the centralized model serves as the gold standard, offering an achievable level of performance. Our objective is to design a method that can closely match centralized performance without requiring all data or features to be aggregated in a single location. We acknowledge that the distributed DIVIDE framework does not consistently outperform centralized models in terms of prediction accuracy. Its primary aim, however, is to enable distributed learning under practical constraints (e.g., data privacy and communication efficiency) and to provide performance as close to a centralized model as possible.

---

### Review · Reviewer_BQdj · 2025-07-30

**Summary Of Contributions:**

The paper introduces DIVIDE, a distributed, hierarchical learning framework for time-series forecasting, focusing on settings where nodes observe only subsets of time-series data. DIVIDE decomposes forecasting models into local models (with flexibility for architecture and data modality) and a global model that fuses local information. The authors provide communication-efficient training algorithms, formal convergence guarantees, and empirical evaluations on few real-world datasets.

Strengths:
* Clear and detailed formulation of the method
* Theoretical contributions
* Communication efficient approach

Weaknesses:
* Privacy was claimed, however it was not supported by empirical or theoretical analysis
* Weak baselines, no federated learning methods compared against
* Certain statements are ''overclaimed"

**Additional Comments:**

Where is the letter V in the name of the approach 'DIVIDE' coming from?

**Audience:**

Yes

**Audience Explanation:**

Many aspects of this work will be interesting to various sub-audiences of TMLR - distributed learning; time-series; theory.

**Broader Impact Concerns:**

No ethical concerns.

**Claims And Evidence:**

Yes

**Claims Explanation:**

Main claims of approach capabilities (other than privacy preservation) appear well supported.

Privacy guarantees are not that strong, as sharing embeddings and gradients could still leak information.

I find some claims a bit of a stretch: "Label sharing might be acceptable for some problems (like classification or regression), since the labels, in many cases, are low-dimensional and might not lead to significant privacy leakage." Even though low dimension - labels like presence of cancer or not in medical applications are quite a significant privacy concern.

" To the best of our knowledge, this is the first time the Markov property of the time series data has been utilized to guarantee convergence of the forecasting algorithms." Even Kalman filter work, from more than half a century ago, relies on Markov property for convergence guarantees.

**Requested Changes:**

Consider adding some stronger baselines like FEDformer, SplitFed, SplitNN.

Soften some of the claims on novelty (e.g. first convergence guarantee using Markov property).

Add discussion on the privacy guarantees.

---

> ### Author Response · Authors · 2025-09-11
> **Response to reviewer BQdj**
>
> **(1) Weak baselines, no federated learning methods compared against.**
> **Response:** We thank the reviewer for pointing out the related works -- FEDformer, SplitFed, SplitNN. We would like to emphasize the differences here.
> &nbsp;&nbsp; FEDformer is a centralized model (here, “FED” refers to Frequency Enhanced Decomposed, not federated). One of its family models, Informer, is already included in our paper. Our goal is to demonstrate that DIVIDE is compatible with various models and that accurate forecasting is achievable in a distributed, privacy-preserving setting, rather than competing with centralized models.
> &nbsp;&nbsp;  SplitNN partitions a single network by layers, while DIVIDE partitions by data sources/modalities, allowing heterogeneous local models and a global fusion model. Importantly, in SplitNN since a single neural network is split among clients and server the backpropagation is performed in a sequential manner at the server and the local clients while for the proposed DIVIDE framework the backpropagation can be done in a parallel fashion. Key differences of DIVIDE with SplitNN include: (i) DIVIDE avoids label sharing by only exchanging embeddings and gradients with provable communication bounds, (ii) DIVIDE enables parallel updates on both the server and the clients while useing independent model architectures and (iii) provide convergence guarantees under non-IID Markov sampling with heterogeneous models—a regime not covered by SplitNN.
> &nbsp;&nbsp;  Empirically, DIVIDE handles multimodal/global tasks with significantly lower communication while matching or surpassing centralized baselines.
>
>
>
>
>
> **(2) Certain statements are ''overclaimed".**
> > **Comment:** I find some claims a bit of a stretch: "Label sharing might be acceptable for some problems (like classification or regression), since the labels, in many cases, are low-dimensional and might not lead to significant privacy leakage." Even though low dimension - labels like presence of cancer or not in medical applications are quite a significant privacy concern.
> > **Comment:** " To the best of our knowledge, this is the first time the Markov property of the time series data has been utilized to guarantee convergence of the forecasting algorithms." Even Kalman filter work, from more than half a century ago, relies on Markov property for convergence guarantees.
>
> **Response:** We appreciate the reviewer’s valuable feedback and agree that the original statements may not have been sufficiently qualified.
> &nbsp;&nbsp;  Regarding the first point, we will revise the text to clarify that the acceptability of label sharing is highly problem-dependent — for instance, labels describing publicly observable, non-personal attributes (e.g., weather conditions, traffic object categories, or device operational states) may carry minimal privacy concerns, whereas labels in sensitive domains such as healthcare can be highly privacy-critical.
> &nbsp;&nbsp;  For the second point, we will reword this claim to properly acknowledge prior work and to more accurately position our contribution within the existing literature.
>
>
>
>
>
> **(3) Privacy was claimed, however it was not supported by empirical or theoretical analysis.**
> > **Comment:** Privacy guarantees are not that strong, as sharing embeddings and gradients could still leak information.
> >
> **Response:** We thank the reviewer for raising this important concern. We agree that sharing embeddings and gradients can, in some cases, pose risks of privacy leakage. However, the developed methods can be readily combined with existing differentially private techniques to provide formal privacy guarantees. In our discussion, our use of the term ‘privacy’ referred to the fact that embeddings (and gradients), being low-dimensional representations generated from heterogeneous models, are comparatively more privacy-preserving than sharing raw data. In the revised manuscript, we will clarify this point and remove any statements that may appear to overstate the privacy guarantees.
>
>
> **(4) Where is the letter V in the name of the approach 'DIVIDE' coming from?**
> **Response:** The full name is Distributed Vertical hIerarchical DEcomposition, where “V” stands for “Vertical.” We will update the manuscript accordingly.

---

### Review · Reviewer_gCmy · 2025-08-26

**Summary Of Contributions:**

This paper introduces DIVIDE, a distributed hierarchical framework for multimodal time series forecasting. Each local node trains its own model on its partial time series (possibly heterogeneous in modality, such as numerical, visual, or sensor data), while a global model aggregates low-dimensional embeddings to capture cross-node correlations and generate both local and global forecasts. The framework supports flexible model selection per node, avoids raw data sharing to preserve privacy, and reduces communication cost by only transmitting embeddings and gradients. The authors provide a theoretical convergence analysis under non-i.i.d. Markovian data assumptions and demonstrate performance gains over centralized and distributed baselines on real-world datasets. A notable empirical finding is that DIVIDE sometimes even surpasses centralized training. The main contributions are: (1) a novel hierarchical decomposition for distributed forecasting; (2) communication- and privacy-efficient training algorithms; (3) convergence guarantees with Markov-dependent samples.

**Audience:**

Yes

**Audience Explanation:**

Yes. TMLR’s audience would be interested in this work, as it addresses an underexplored but practically important problem: distributed multimodal time series forecasting. The hierarchical structure, theoretical guarantees, and multimodal support make the paper relevant to both machine learning theory and applied communities (federated learning, IoT, and time series analysis).

**Broader Impact Concerns:**

No major ethical concerns are identified. The framework respects privacy by avoiding raw data sharing and is well aligned with responsible AI practices.

**Claims And Evidence:**

Yes

**Claims Explanation:**

Yes. The claims are generally supported with solid theoretical and empirical evidence. The convergence proof is clearly derived under Markovian assumptions, and experiments show DIVIDE outperforming centralized and distributed baselines. However, the explanation of why DIVIDE occasionally surpasses centralized models (p. 8) remains anecdotal and would benefit from deeper analysis or ablation.

**Requested Changes:**

1. Clarify empirical observations: On p. 8, provide more concrete explanations or controlled experiments on why DIVIDE outperforms centralized models. For example, analyze whether hierarchical modeling reduces overfitting or whether data heterogeneity benefits distributed aggregation.

2. Broader multimodal results): In Section 5.4, the multimodal evaluation with DeepSense6G is promising, but more detail on local model choices per modality should be given for clarity.

3. Communication overhead analysis:The communication cost discussion is somewhat abstract. Providing quantitative comparisons would strengthen the claim of communication efficiency.

---

> ### Author Response · Authors · 2025-09-11
> **Response to reviewer gCmy**
>
> **(1) Clarify empirical observations: On p. 8, provide more concrete explanations or controlled experiments on why DIVIDE outperforms centralized models. For example, analyze whether hierarchical modeling reduces overfitting or whether data heterogeneity benefits distributed aggregation.**
> **Response:** We appreciate the valuable feedback from the reviewer. In the revised version, we will provide a more detailed explanation in Section 5.3 to elucidate the potential reasons why DIVIDE may outperform centralized models in some experiments. We note that existing experimental results (such as Table 1 and Figure 2) indicate that the proposed hierarchical model may improve generalization performance. We attribute this performance gain to DIVIDE’s ability to efficiently leverage correlations across nodes. This is further supported by the observation that models incorporating a global component consistently outperform those without, suggesting that aggregation mitigates overfitting to local patterns. Moreover, as discussed in Sections 3 and 4, DIVIDE is particularly advantageous under data heterogeneity, where embeddings from different modalities provide complementary information that centralized models often overfit and fail to fully exploit.
>
>
>
>
> **(2) Broader multimodal results: In Section 5.4, the multimodal evaluation with DeepSense6G is promising, but more detail on local model choices per modality should be given for clarity.**
> **Response:** We thank the reviewer for the suggestion. In the revised version, we will supplement Appendix B.3 with details on local model selection: the choice of local models primarily depends on the type of local data modality and task type. Specifically, we currently use CNNs for video streams, PointNet for point cloud data processing, and RNNs/LSTMs for time-series data processing.
>
>
>
> **(3) Communication overhead analysis:The communication cost discussion is somewhat abstract. Providing quantitative comparisons would strengthen the claim of communication efficiency.**
> **Response:** We thank the reviewer for the insightful suggestion. In Remark 3 of Section 3.3, we will add the communication cost comparison. In short, DIVIDE only transmits low-dimensional embeddings and gradients per round, with a complexity of $O(Kd)$, where $K$ is the number of local models and $d$ is the embedding dimension. In contrast, FedAvg and its variants (FedProx, FedDyn, FedNova) require synchronizing the entire model, with a complexity of $O(KP)$, where $P$ is the total number of model parameters, we typically have $d≪P$. Other approaches such as Quantized FL (QFedAvg, SignSGD, TernGrad), Sparsified FL, or Low-rank/Sketching reduce communication only by constant factors but increase the variance of the computed gradients and still fundamentally depend on the parameter size.

---

### Decision · Action_Editor_ytQk · 2025-10-10

**Recommendation:** Accept with minor revision

**Additional Comments:**

This paper addresses an important and timely problem and demonstrates solid effort in both theoretical motivation and empirical validation. The presentation is clear, the experiments are well executed, and the topic is of genuine interest to the TMLR audience working on distributed and privacy-preserving learning.

Overall, the results support the authors’ main claims, though several aspects would benefit from further clarification and minor strengthening to improve completeness and interpretability. In particular, providing a more detailed analysis of why DIVIDE sometimes surpasses centralized baselines could enhance understanding of its behavior and boundaries. Likewise, clarifying the distinctions from related designs such as fully connected GNNs or latent master-node architectures would help position the method more clearly within existing literature.

Additional analyses of privacy guarantees and inclusion of one or two stronger or more comparable baselines would further solidify the empirical evidence. Finally, since the paper currently uses the term “novel,” the authors might either remove it or provide a short contextual note referencing related approaches to avoid possible ambiguity.

With these modest revisions, the paper would constitute a convincing and well-rounded contribution suitable for publication in TMLR.

**Audience:**

Yes

**Audience Explanation:**

The researches working on multimodal time series analysis will be interested in this topic.

**Claims And Evidence:**

No

**Claims Explanation:**

The paper presents a well-structured hierarchical framework for distributed multimodal time series forecasting and provides theoretical convergence analysis under Markovian assumptions. Some novelty statements are overstated given prior work. Empirical results show DIVIDE occasionally matching but not consistently surpassing centralized baselines.